# Impaired respiration elicits SrrAB-dependent programmed cell lysis and biofilm formation in *Staphylococcus aureus*

Ameya A Mashruwala, Adriana van de Guchte, Jeffrey M Boyd*

Department of Biochemistry and Microbiology, Rutgers University, New Brunswick, United States

**Abstract** Biofilms are communities of microorganisms attached to a surface or each other. Biofilm-associated cells are the etiologic agents of recurrent *Staphylococcus aureus* infections. Infected human tissues are hypoxic or anoxic. *S. aureus* increases biofilm formation in response to hypoxia, but how this occurs is unknown. In the current study we report that oxygen influences biofilm formation in its capacity as a terminal electron acceptor for cellular respiration. Genetic, physiological, or chemical inhibition of respiratory processes elicited increased biofilm formation. Impaired respiration led to increased cell lysis via divergent regulation of two processes: increased expression of the AtlA murein hydrolase and decreased expression of wall-teichoic acids. The AltA-dependent release of cytosolic DNA contributed to increased biofilm formation. Further, cell lysis and biofilm formation were governed by the SrrAB two-component regulatory system. Data presented support a model wherein SrrAB-dependent biofilm formation occurs in response to the accumulation of reduced menaquinone.

*For correspondence: jmboyd@ SEBS.rutgers.edu

**Competing interests:** The authors declare that no competing interests exist.

## Introduction

*Staphylococcus aureus* is a commensal bacterium that is estimated to colonize between 20–50% of the healthy human population (*Naimi et al., 2003*; *Graham et al., 2006*; *Enright et al., 2002*; *Ohara-Nemoto et al., 2008*; *Zafar et al., 2007*). Colonization typically occurs in the nares, throat, or on the skin (*Ohara-Nemoto et al., 2008*; *Zafar et al., 2007*; *Hamdan-Partida et al., 2010*). Under select conditions, *S. aureus* is capable of causing both invasive as well as non-invasive infections (*Klevens et al., 2007*; *Tong et al., 2015*; *Williamson et al., 2013*). The dominant fraction of invasive infections caused by this bacterium occur in the context of bacteremia (*Klevens et al., 2007*). In addition, *S. aureus* can infect and cause diseases of the lungs (penumonia), skin (cellulitis), skeletal tissues (ostoemyelitis), and heart tissue (endocarditis), as well as septic shock (*Klevens et al., 2007*; *Tong et al., 2015*). In the United States, pneumonia and septic shock are rapidly progressing infections and are often fatal with mortality rates in the United States (US) of 30–55% (*Klevens et al., 2007*). While bacteremia and endocarditis infections have a lower degree of mortality, they are associated with a higher degree of recurrence, suggestive of therapeutic recalcitrance (*Klevens et al., 2007*). A recent epidemiological analysis of ~8,700 cases of invasive *S. aureus* infections in the US found that nearly 92% cases required hospitalization (*Klevens et al., 2007*).

Historically, *S. aureus* infections in the US were largely nosocomial in origin; however, their onset or occurrence increasingly transpires in community settings (*Klevens et al., 2007*; *Tenover et al., 2006*). In the United States, pulsed-field type USA300 methicillin-resistant *S. aureus* (MRSA) has emerged as the dominant etiologic agent of community-associated invasive infections

**eLife digest** Millions of bacteria live on the human body. Generally these bacteria co-exist with us peacefully, but sometimes certain bacteria may enter the body and cause infections, such as gum disease or a bone infection called osteomyelitis. Many of these infections are thought to occur when the bacteria become able to form complex communities called biofilms. Bacteria living in a biofilm cooperate and make lifestyle choices as a community, so in this way, they behave like a single organism containing many cells.

A sticky glue-like material called the matrix holds the bacteria in a biofilm together. This matrix protects the bacteria in the biofilm from both the human immune system and antibiotics, allowing infections to develop and making them difficult to treat.

Previous research has shown that the supply and level of oxygen in infected tissues decreases as an infection gets worse. One bacterium that typically lives peacefully on our bodies, called *Staphylococcus aureus,* can sometimes cause serious biofilm-associated infections. *S. aureus* forms biofilms more readily when oxygen is in short supply, but it was not known how these biofilms form. Understanding how *S. aureus* forms biofilms could help scientists develop better treatments for bacterial infections.

Most bacterial cells have a cell wall to provide them with structural support. Mashruwala *et al.* found that, when oxygen levels are low, *S. aureus* decreases the production of a type of sugar that makes up the cell wall. At the same time, the bacteria produce more of an enzyme that breaks down cell walls. Together, these processes cause some of the bacteria cells to break open. The contents of these broken cells, including their DNA, help form the matrix that will hold together and protect the other bacterial cells in the biofilm. The experiments also identified a protein called SrrAB that switches on the process that ruptures the cells when oxygen is low.

The findings of Mashruwala *et al.* show how bacteria grown in the laboratory form biofilms when they are starved of oxygen. The next steps following on from this work are to find out whether the same thing happens when bacteria infect animals and whether drugs that block the rupturing of bacterial cells could be used to treat infections.

(*Klevens et al., 2007*). Treatment of *S. aureus* infections is often problematic due to the increasing prevalence of antibiotic resistance. *S. aureus* strains have been isolated that are resistant to nearly all clinically available antibiotics, including the last-line antibiotics linezolid and daptomycin (*Sass et al., 2012*; *Sánchez García et al., 2010*).

Biofilms are architecturally complex, multicellular communities of microorganisms of either mono- or poly-microbial compositions (*Costerton, 1995*; *Costerton et al., 1995*). It has been theorized, based upon studies using direct techniques, such as microscopy, that ~99% of bacteria establish biofilms in their natural environments (*Costerton et al., 1995*). A number of persistent and chronic infections in humans, such as periodontis and cystic fibrosis, are associated with the ability of the microorganisms to establish biofilms (*Sedghizadeh et al., 2009*; *Costerton et al., 1999*). In addition, biofilms of infectious agents are well characterized to form upon biomedical devices such as prosthetics, heart valves, catheters, and contact lenses (*Costerton et al., 1999*, *2005*; *Bispo and Haas, 2015*). A number of staphylococcal infections, such as osteomyelitis, are also intimately connected to the ability of the bacterium to form biofilms (*Joo and Otto, 2012*; *Otto, 2008*). Reflective of their clinical significance, biofilms are considered to be the etiologic agents of recurrent staphylococcal infections (*Joo and Otto, 2012*; *Otto, 2008*).

*S. aureus* biofilms are typically composed of one or more extracellular polymeric molecules (DNA, proteins, or polysaccharides) that provide structural integrity and may also facilitate intercellular adhesion (*Rice et al., 2007*; *Schwartz et al., 2012*; *Boles and Horswill, 2008*; *Cramton et al., 1999*). The polymers interact to facilitate the formation an extracellular matrix. This matrix provides protection from environmental stress, innate immunity, as well as therapeutic agents (*Davies, 2003*). The polymer(s) utilized to facilitate biofilm formation can vary between staphylococcal isolates with some favoring DNA and/or proteins and others polysaccharides (*Rice et al., 2007*; *Schwartz et al., 2012*; *Boles and Horswill, 2008*; *Cramton et al., 1999*). The complexity of biofilm formation results

in this process being highly regulated and deterministic. Biofilm formation in *S. aureus* is responsive to diverse signals including nutrient limitation and quorum sensing (*Joo and Otto, 2012*; *Otto, 2008*; *Boles and Horswill, 2008*; *Majerczyk et al., 2008*).

Oxygen concentrations vary greatly between healthy human tissues (between 19.7 to ~1.5%; normoxia) (*Carreau et al., 2011*). Oxygen concentrations also vary between healthy and infected or necrotic tissues, as well as in wounds, where concentrations are estimated to be below 1% (hypoxic) or anoxic (*Carreau et al., 2011*; *Vogelberg and König, 1993*; *Arnold et al., 1987*). A recent study found that *S. aureus* infections in skeletal tissues (osteomyelitis) cause an ~3 fold decrease in oxygen concentrations resulting in increasing hypoxia as infection proceeds (*Wilde et al., 2015*).

Multiple studies have focused upon the human systems that are active under hypoxia or anoxia and aid in combating bacterial infections. However, relatively little is known about how *S. aureus* mount a response to hypoxia or anoxia. A study by Cramton *et al.* found that decreased oxygen concentrations result in increased biofilm formation in *S. aureus* (*Cramton et al., 2001*). An alternate study found that *S. aureus* growing in biofilms are starved for oxygen and that the rate of oxygen depletion is proportional to the rate of biofilm maturation (*Zhu et al., 2007*). Cramton *et al.* also found that decreased oxygen concentrations lead to increased production of the polysaccharide intercellular adhesin (PIA), which is a polymer used by some *S. aureus* isolates to facilitate intercellular adhesion (*Cramton et al., 2001*). However, the role or requirement of PIA in low oxygen biofilms is unclear since biofilm formation in a PIA deficient strain was not examined (*Cramton et al., 2001*). It is also unclear how the lack of oxygen, a cell permeable molecule, translates into increased biofilm formation.

Two-component regulatory systems (TCRS) are modular signal transduction pathways that facilitate the integration of multiple stimuli into cellular signaling circuits, allowing for a rapid and robust response to environmental alterations (*Stock et al., 2000*; *Stephenson and Hoch, 2002*). In *S. aureus*, which encodes for classical TCRS, the systems are predicted to be composed of a histidine kinase (HK) and a DNA-binding response regulator (RR). The HK interacts with the environmental stimulus and can be either membrane associated or cytosolic. Upon stimulation, the HK alters the levels of the phosphoryl group upon the RR. In the case of most (but not all) DNA-binding RRs, altered phosphoryl levels modify the affinity of the RR for DNA resulting in altered gene transcription and a tailored physiological response (*Stock et al., 2000*; *Stephenson and Hoch, 2002*).

The goal of this study was to examine the mechanisms by which oxygen affects *S. aureus* biofilm formation. Data presented show that oxygen impacts biofilm formation in its capacity as a terminal electron acceptor in cellular respiration. Consequently, growth conditions that diminish respiration elicit increased biofilm formation. Impaired respiration leads to increased cell lysis via increased expression of the AltA murein hydrolase and a concomitant decrease in the expression of wall-teichoic acids. The regulatory tuning of these two processes in a divergent manner affects cell lysis. Increased biofilm formation and cell lysis is a programmed mechanism that is governed by the SrrAB TCRS. Genetic evidence suggests that SrrAB-dependent biofilm formation occurs in response to the accumulation of reduced menaquinone.

## Results

### *S. aureus* forms robust biofilms in the absence of oxygen

The influence of anaerobiosis upon biofilm formation of *S. aureus* was examined. Regulatory networks integral to staphylococcal physiology differ between *S. aureus* isolates (*Herbert et al., 2010*; *Memmi et al., 2012*). Biofilm formation was examined in diverse *S. aureus* isolates that vary in their ability to form biofilms (LAC, SH1000, MW2, N315). Strains were cultured aerobically, with a seal that allows free diffusion of gases, or anaerobically (in a COY anaerobic chamber equipped with an oxygen scavenging catalyst, $O_2$ <1 ppm) prior to quantifying biofilms. Biofilm formation increased substantially for each strain during anaerobic growth (between ~4–30 fold) (*Figure 1A and B*). Unless specifically mentioned, the experiments described henceforth were conducted using the community-associated MRSA strain LAC (hereafter wild-type; WT).

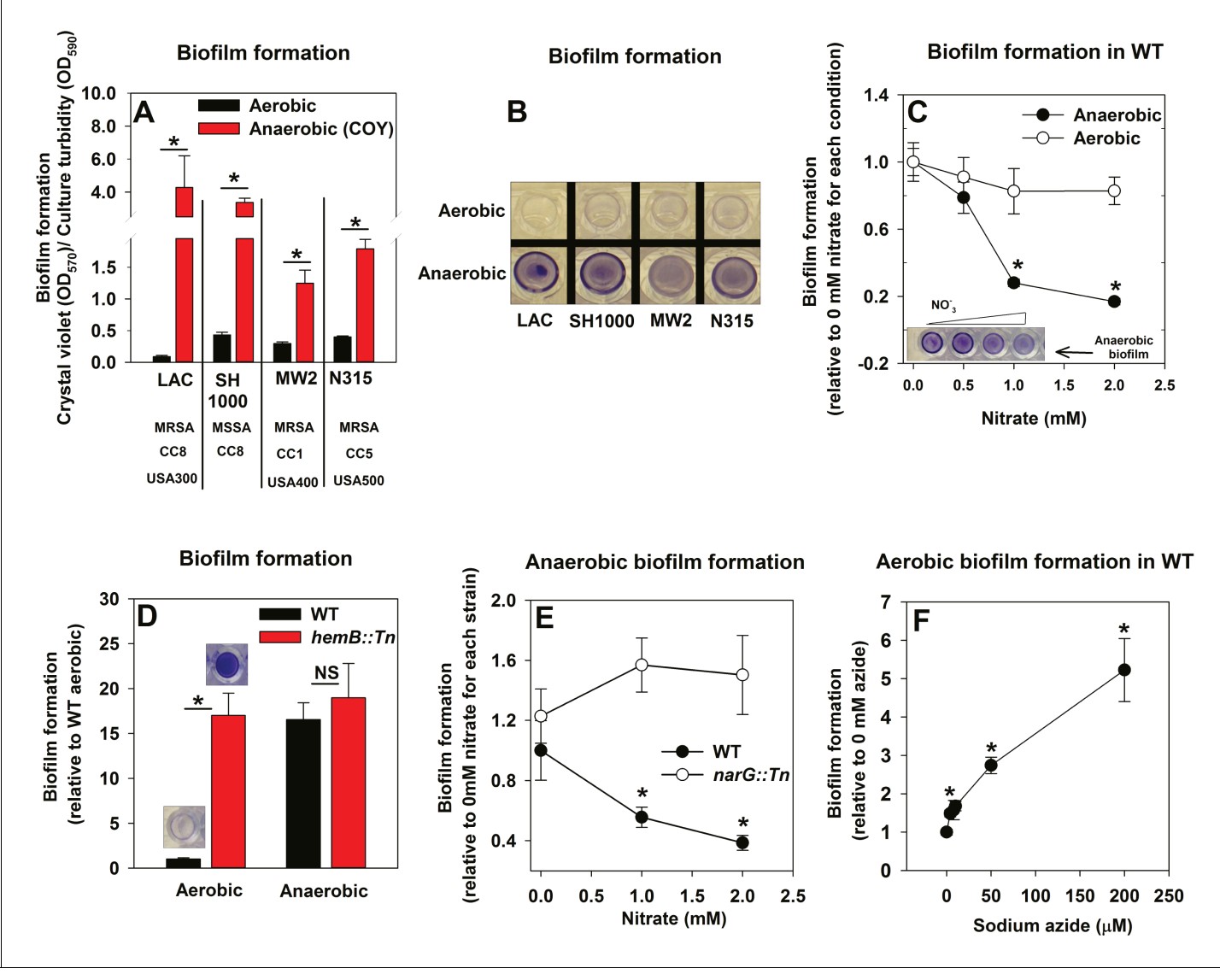

**Figure 1.** Oxygen impacts biofilm formation in its capacity as a terminal electron acceptor. Panels A and B; Anaerobic growth elicits increased biofilm formation in multiple *S. aureus* isolates. Biofilm formation of the LAC (JMB1100; hereafter wild-type (WT)), SH1000 (JMB 1323), MW2 (JMB1324) and N315 (JMB 7570) isolates following aerobic or anaerobic growth is displayed. MRSA denotes methicillin resistance, MSSA denotes methicillin sensitivity, CC denotes clonal complex type and the USA number denotes the pulsed-field gel electrophoeresis type. Panel C; Supplementing growth media with the alternate terminal electron acceptor nitrate results in decreased biofilm formation during anaerobic growth. Biofilm formation for WT following aerobic or anaerobic growth and in media containing between 0–2 mM sodium nitrate is displayed. Panel D; A strain incapable of respiration upon oxygen forms increased biofilms when cultured aerobically, but not fermentatively. Biofilm formation for the WT and *hemB::Tn* (JMB6037) strains following aerobic or anaerobic growth is displayed. Panel E; Nitrate supplementation does not decrease anaerobic biofilm formation in a nitrate reductase mutant. Biofilm formation for the WT and *narG::Tn* (JMB7277) strains following anaerobic growth and in media containing between 0–2 mM sodium nitrate. Panel F; Chemical inhibition of respiration elicits increased biofilm formation during aerobic growth. Biofilm formation for the WT following aerobic growth in media supplemented with 0–250 µM sodium azide. The data represent the average values of eight wells (Panels A, C-E) or quadruplicates (Panel F) and error bars represent standard deviations. Representative photographs of biofilms formed upon the surface of a 96-well microtiter plate and stained with crystal violet are displayed in Panel B or insets in Panel C and D. Error bars are displayed for all data, but on occasion may be too small to see. Statistical significance was calculated using a two-tail Student's t-test and p-values>0.05 were considered to be not significant while * indicates p-value of <0.05.

## Oxygen influences biofilm formation in its capacity as a terminal electron acceptor for cellular respiration

The principal influence of oxygen upon staphylococcal physiology is achieved in its capacity as a terminal electron acceptor (TEA) for respiration. Increased biofilm formation during anaerobic growth occurred upon culture in a medium lacking a terminal electron acceptor (fermentative growth). We tested the hypothesis that impaired respiration is a signal that elicits biofilm formation. In addition to oxygen, *S. aureus* can utilize nitrate as a TEA. Anaerobic biofilm formation decreased, as the concentration of nitrate provided in the medium was increased (*Figure 1C*). The addition of nitrate to aerobic cultures did not significantly alter biofilm formation (*Figure 1C*).

We reasoned that strains incapable of respiration would display increased biofilm formation. Heme auxotrophs have non-functional terminal oxidases and are unable to respire. They, form small colonies when cultured in the presence of oxygen, and therefore are termed small-colony variants (*Hammer et al., 2013*). A *hemB::Tn* strain formed considerably more biofilm than the WT when cultured aerobically, but displayed biofilm formation similar to the WT when cultured fermentatively (*Figure 1D*). Likewise, nitrate supplementation did not decrease anaerobic biofilm formation in a nitrate reductase (*narG::Tn*) mutant, which is unable to utilize nitrate as a TEA (*Figure 1E*) (*Schlag et al., 2008*; *Burke and Lascelles, 1975*). To further test our premise, biofilm formation was examined in the WT cultured aerobically with varying amounts of the respiratory poison sodium azide. Biofilm formation increased in synchrony with the concentration of sodium azide in the growth medium (*Figure 1F*).

From *Figure 1* we concluded that decreased cellular respiration results in increased biofilm formation. Further, biofilm formation was responsive to the concentration of a terminal electron acceptor or the ability of cells to respire.

## Impaired respiration leads to AtlA-dependent release of DNA and cytosolic proteins facilitating biofilm formation

We sought to understand the mechanisms underlying the formation of fermentative biofilms. We examined the dependence of fermentative biofilms upon one or more of the described structural polymers: intercellular polysaccharide adhesin (PIA), high-molecular weight extracellular DNA (eDNA), or proteins (*Rice et al., 2007*; *Schwartz et al., 2012*; *Boles and Horswill, 2008*; *Cramton et al., 1999*; *Foulston et al., 2014*). The *icaABCD* operon encodes for proteins required to biosynthesize PIA (*Cramton et al., 1999*). Strains lacking functional IcaA, IcaB, or IcaC were not attenuated in fermentative biofilm formation, suggesting that PIA is dispensable for this phenotype (*Figure 2—figure supplement 1*). However, supplementation of the growth medium with DNase, which degrades DNA, substantially attenuated biofilm formation suggesting that DNA is an integral component of fermentative biofilms (*Figure 2A*). Consistent with this theory, the accumulation of high-molecular weight extracellular DNA (eDNA) increased appreciably in the matrix of fermenting biofilms (*Figure 2B*).

Prevailing models suggest that eDNA in staphylococcal biofilms arises as a consequence of a self digestive cell-lysis process (autolysis), which results in the release of high-molecular weight genomic DNA (*Rice et al., 2007*; *Foulston et al., 2014*). Polyanethole sulfonate (PAS) inhibits *S. aureus* autolysis (*Wecke et al., 1986*; *Yabu and Kaneda, 1995*). Supplementing growth media with PAS diminished fermentative biofilm formation (*Figure 2—figure supplement 2*).

Peptidoglycan (murein) cleavage would be necessary for autolysis. The *S. aureus* genome encodes for multiple murein hydrolases (*Navarre et al., 1999*; *Frankel et al., 2011*). Fermentative biofilm formation was examined in a set of strains that each lacked one predicted murein hydrolase. One strain, with a disruption in the gene encoding for the AtlA murein hydrolase (*atlA::Tn*), was attenuated in biofilm formation (*Figure 2—figure supplement 3*). AtlA has been previously implicated to be required for biofilm formation during aerobic growth (*Bose et al., 2012*; *Houston et al., 2011*; *Biswas et al., 2006*). The *atlA::Tn* strain displayed decreased biofilm formation in the presence of oxygen (~1 fold decrease) and this phenotype was exacerbated (~10 fold decrease) in fermenting cultures (*Figure 2C*) suggesting that the role of AtlA in biofilm formation is increased during fermentative growth. Moreover, eDNA accumulation was greatly decreased in the biofilm matrix of the fermentatively cultured *atlA::Tn* strain (*Figure 2D*).

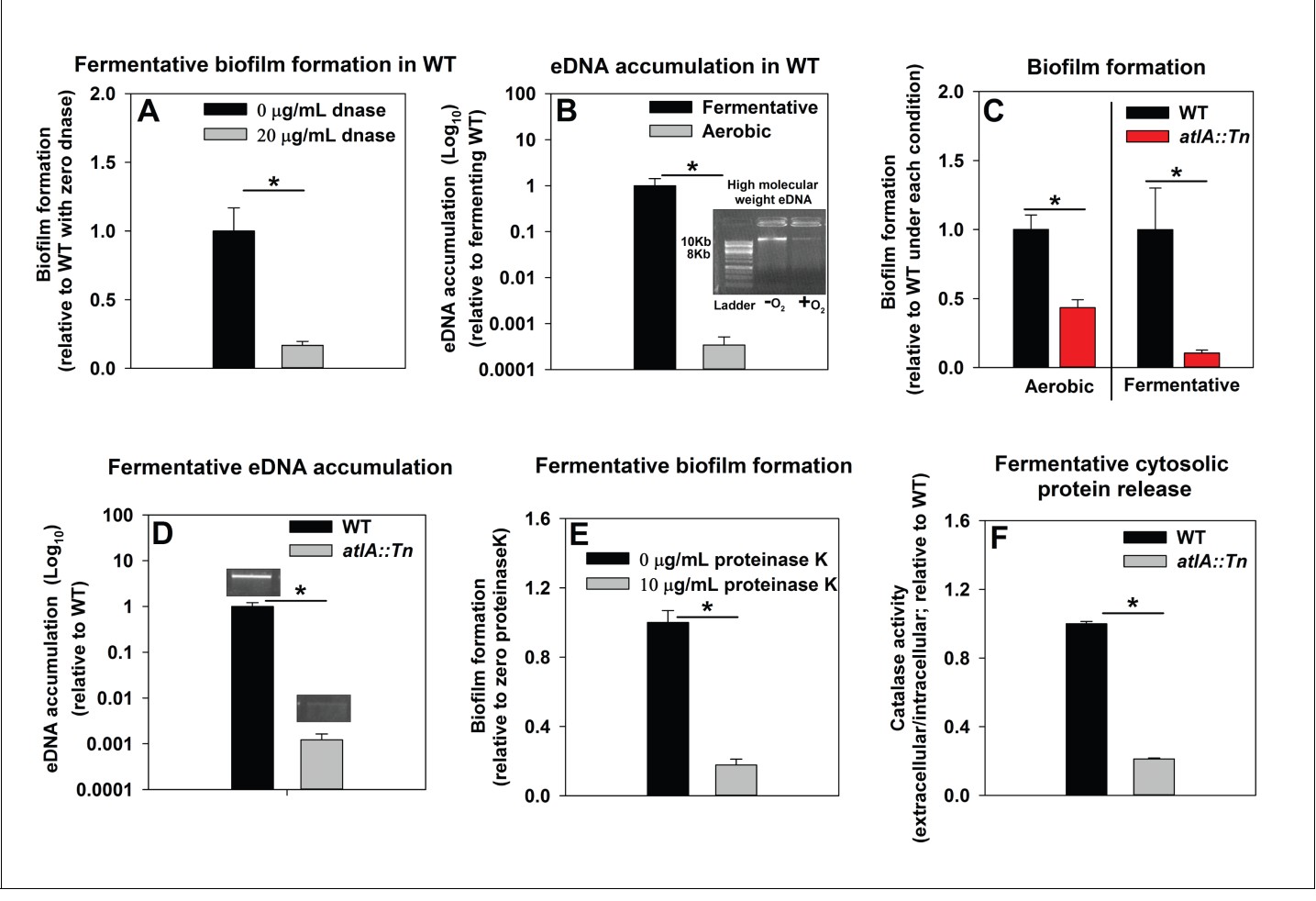

**Figure 2.** Impaired respiration results in AtlA-dependent release of high-molecular weight DNA, cytoplasmic proteins and an increase in biofilm formation. Panel A; Fermentative biofilm formation is attenuated upon supplementation of growth medium with DNase. Biofilm formation of the WT (JMB 1100) following fermentative growth in media with or without 20 μg/mL DNase is displayed. Panel B; High-molecular weight DNA (eDNA) accumulation is increased in the biofilm matrix of fermenting cells. Biofilms of the WT were cultured aerobically or fermentatively, eDNA was extracted, and analyzed using agarose gel electrophoeresis (inset photograph). The data were normalized to the viable cell count, and thereafter, to eDNA accumulation in fermenting WT. Panel C; Fermentative biofilm formation is dependent upon the AtlA murein hydrolase. Biofilm formation for the WT and the *atlA::Tn* (JMB 6625) strains cultured aerobically or fermentatively is displayed. Panel D; eDNA accumulation in fermenting biofilms is dependent upon AtlA. Biofilms of the WT and *atlA::Tn* strains were cultured fermentatively and eDNA accumulation assessed. The data were normalized to the viable cell count, and thereafter, to eDNA accumulation in WT. Panel E; Fermentative biofilm formation is attenuated upon supplementation of growth medium with Proteinase K. Biofilm formation for the WT following fermentative growth in media with or without 10 μg/mL Proteinase K is displayed. Panel F; Fermentative growth results in AtlA-dependent release of a cytosolic protein into the extracellular milleu. Biofilms of the WT and *atlA::Tn* strains were cultured fermentatively and the activity of the cytosolic protein catalase (Kat) was measured in the spent media supernatant. The data were normalized to intracellular Kat activity, and thereafter to WT levels. The data represent the average values of eight wells (Panels A, C and E), sextuplets (Panel B) or triplicates (Panels D and F) and error bars represent standard deviations. Representative photographs of high-molecular weight eDNA are displayed in Panel B or inset in Panel D. Error bars are displayed for all data, but might be too small to see on occasion. Statistical significance was calculated using a two-tail Student's t-test and p-values>0.05 were considered to be not significant while * indicates p-value of <0.05.

The following figure supplements are available for figure 2:

**Figure supplement 1.** Polysaccharide intercellular adhesin (PIA) is dispensable for fermentative biofilm formation.

**Figure supplement 2.** Supplementing growth media with the autolysis inhibitor polyanethole sulfonate (PAS) attenuates fermentative biofilm formation.

**Figure supplement 3.** Fermentative biofilm formation is dependent on the AtlA murein hydrolase.

*Figure 2 continued on next page*

*Figure 2 continued*

**Figure supplement 4.** Cytosolic protein release is increased upon fermentative growth.

A recent study found that cytosolic proteins form a significant portion of staphylococcal biofilm matrixes (*Foulston et al., 2014*). AtlA has been implicated in the release of cytosolic proteins into the extracellular milleu (*Pasztor et al., 2010*). The supplementation of media with proteinase K, which degrades proteins, attenuated fermentative biofilm formation, suggesting that in addition to eDNA, proteins also form an integral part of the biofilm matrix in fermenting cells (*Figure 2E*). To further examine this, the activity of catalase (Kat) (*Cosgrove et al., 2007*; *Mashruwala et al., 2016a*), an abundant intracellular protein (*Cosgrove et al., 2007*), was measured in the spent media supernatants. The spent media supernatant from fermenting WT had ~5 fold increased Kat activity relative to aerobically cultured WT (*Figure 2—figure supplement 4*). Kat activity was decreased by ~5 fold in the spent media supernatant from the fermentatively cultured *altA::Tn* strain (*Figure 2F*). These data were normalized to intracellular Kat activity to negate for potential changes in Kat expression.

From *Figure 2* and *Figure 2—figure supplements 1–4* we concluded that fermenting cells release an increased quantity of DNA and cytoplasmic proteins, into their extracellular mileu, in an AtlA-dependent manner. The eDNA and proteins are incorporated into the biofilm matrix and contribute to biofilm formation.

## Impaired respiration elicits increased expression of AtlA and alterations that make cells more amenable to cleavage by AtlA

Three scenarios could underlie the increased role of AtlA in fermentative biofilm formation. First, the expression of AtlA is increased leading to increased autolysis. Second, cell walls are altered in order to make them more amenable to AtlA-dependent lysis. Third, a combination of scenarios one and two. To discern which of these scenarios is operative in fermenting cells, the abundance of the *atlA* transcript was assessed in WT cultured aerobically or fermentatively. The *atlA* transcript was increased ~5 fold upon fermentative culture (*Figure 3A*). Subsequently, AtlA activity was examined within the context of intact whole cells using autolysis assays (*Bose et al., 2012*). Fermentatively cultured WT cells underwent autolysis faster than those cultured aerobically. The *atlA::Tn* strain, cultured aerobically or fermentatively, was severely deficient in undergoing autolysis suggesting that AtlA was the dominant murein hydrolase contributing to autolysis under the growth conditions examined (*Figure 3B*).

Murein hydrolase assays were used to quantify AtlA-dependent bacteriolytic activity. The WT and *atlA::Tn* strains were cultured aerobically or fermentatively, cell-wall associated proteins were detached (hereafter CW-extracts), and bacteriolytic activity was examined using heat-killed *Micrococcus luteus* as a substrate. CW-extracts from fermenting WT lysed *M. luteus* more rapidly than CW-extracts from WT cultured aerobically (*Figure 3C*). Bacteriolytic activity was nearly undetectable when using CW-extracts from the *atlA::Tn* strain cultured aerobically or fermentatively. These data confirmed that AtlA was the dominant murein hydrolase in the extracts and increased AtlA activity was associated with the WT cultured fermentatively (*Figure 3C*).

We next examined whether cell walls were altered in order to make them more amenable to AtlA. The WT strain was cultured aerobically or fermentatively, heat-killed to inactivate native autolysins, and the cells were subsequently provided as substrates in murein hydrolase assays. AtlA is a bifunctional enzyme that is proteolytically cleaved into a N-acetylmuramyl-L-alanine amidase (AM) and endo-$\beta$-N-acetylglucosaminidase (GL) ((*Oshida et al., 1995*) and illustrated in *Figure 3—figure supplement 1*). The use of *M. luteus* and *S. aureus* cells as substrates allows for differentiation between AM and GL activities (*Oshida et al., 1995*; *Wadström and Hisatsune, 1970*). GL displays poor activity against *S. aureus*, but is capable of cleaving *M. luteus*. Murein hydrolase assays were conducted using CW-extracts obtained from a *ΔatlA* strain carrying empty vector or plasmids encoding for full length AtlA (p*atlA*), AM only (p*atlA*$_{AM}$) and GL only (p*atlA*$_{GL}$) (*Bose et al., 2012*). Lysis of heat-killed *S. aureus*, as well as *M. luteus*, was undetectable with CW-extracts from the *ΔatlA* strain carrying empty vector verifying that bacteriolytic activity under the conditions examined was

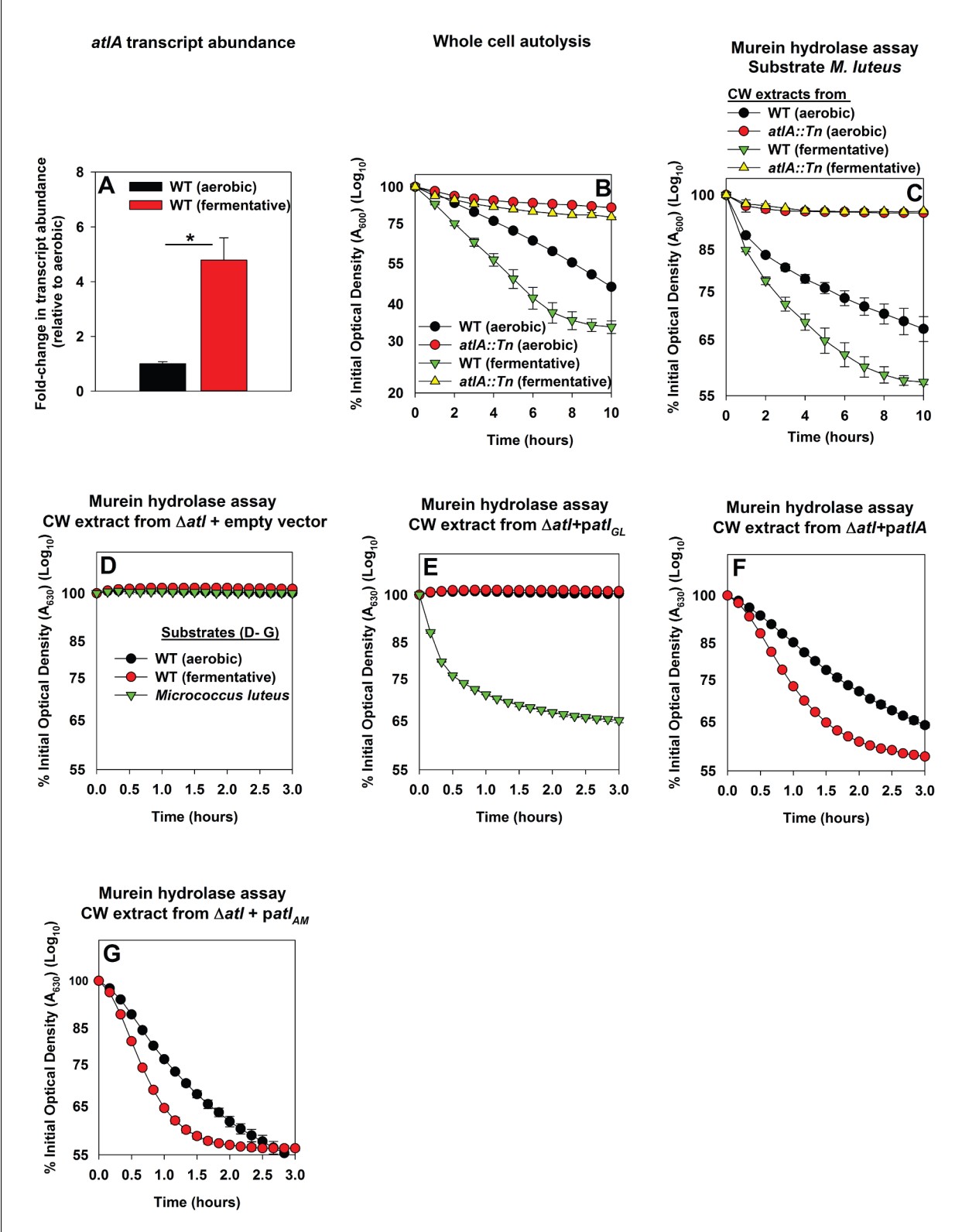

**Figure 3.** Impaired respiration elicits increased expression of AtlA and alterations that make cells more amenable to cleavage by AtlA. Panel A; The *atlA* transcript is increased upon fermentative growth. Biofilms of the WT (JMB 1100) were cultured aerobically or fermentatively, mRNA was extracted, and the abundance of the *atlA* transcript was quantified. The data were normalized to 16S rRNA levels, and thereafter, to levels observed aerobically. Panel B; Fermenting cells undergo increased autolysis in an AtlA-dependent manner. The WT and *atlA::Tn* (JMB 6625) strains were cultured aerobically

*Figure 3 continued on next page*

*Figure 3 continued*
or fermentatively and autolysis was examined in intact whole cells. Panel C; AtlA-dependent bacteriolytic activity is increased in fermenting cells. Murein-hydrolase activity in cell-wall associated proteins (CW-extracts) detached from the WT or *atlA::Tn* strains cultured aerobically or fermentatively is displayed (pH of 7.5). Heat-killed *Micrococcus luteus* was used as a substrate. Panel D-G; Fermenting cells are more amenable to AtlA and N-acetylmuramyl-L-alanine amidase (AM)-dependent cleavage. Murein-hydrolase activity using CW-extracts detached from a *ΔatlA* strain (KB 5000) carrying plasmids encoding for empty vector control (Panel D), GL only (p*atlA*$_{GL}$) (Panel E), full-length AtlA (p*atlA*) (Panel F), or AM only (p*atlA*$_{AM}$) (Panel G) upon heat-killed cells of the WT cultured aerobically or fermentatively or *M. luteus* as substrates is displayed (pH of 7.5). The data in Panel A represent the average values of triplicates. Statistical significance was calculated using a two-tail Student's t-test and * indicates p-value of <0.05. The data in Panels B-G represent the average value of technical duplicates from one set of substrate preparation, autolysis experiments, or CW extract preparations. Autolysis experiments or the preparation of heat-killed substrates or CW-extracts were conducted on least three separate occasions and similar results were obtained. Error bars in all panels represent standard deviations. Error bars are displayed for all data, but might be too small to see on occasion.
The following figure supplement is available for figure 3:

**Figure supplement 1.** Representation of the full -length AtlA precursor protein and of the plasmid encoded variants used in this study.

dependent upon AtlA, AM, or GL (*Figure 3D*). CW-extracts from the *ΔatlA* strain carrying p*atlA*$_{GL}$ did not lyse *S. aureus*, but proficiently lysed *M. luteus*, confirming that *S. aureus* are poor substrates for GL (*Figure 3E*). CW-extracts from the *ΔatlA* strain carrying p*atlA* or p*atlA*$_{AM}$ lysed fermentatively cultured heat-killed WT at a faster rate than aerobically cultured WT, suggesting fermenting *S. aureus* cells are more amenable to cleavage by AtlA and AM (*Figure 3F–G*).

## Decreased expression of wall-teichoic acids in fermenting cells increases their amenability towards cleavage by AtlA

Wall-teichoic acids (WTA) are cell-surface glycopolymers that are covalently attached to peptidoglycan. The biosynthetic pathway for WTA in *S. aureus* is illustrated in *Figure 4A*. WTA negatively modulate AtlA activity (*Biswas et al., 2012*; *Schlag et al., 2010*). Decreased expression of WTA during fermentative growth could result in cells that are more amenable to AtlA-dependent lysis. Consistent with this logic, the transcription of genes encoding for proteins in the WTA biosynthetic pathway (*tarA*, *tarO*, *tarB*, *tarH*) was decreased during fermentative growth (between 6–50 fold) (*Figure 4B*).

Tunicamycin is an inhibitor of TarO and MnaA, which are necessary for WTA biosynthesis (*Figure 4A*) (*Campbell et al., 2011*; *Mann et al., 2016*; *Hancock et al., 1976*). *S. aureus* cultured in the presence of tunicamycin do not synthesize WTA (*Campbell et al., 2011*). WT was cultured aerobically or fermentatively in the presence or absence of tunicamycin, the cells were heat-killed, and used as substrates in murein hydrolase assays. WT cells cultured aerobically with tunicamycin were lysed at a rate similar to that of fermentatively cultured cells by CW-extracts from a *ΔatlA* strain carrying either p*atlA*$_{AM}$ or p*atlA*. This confirmed that changes in WTA expression alter the amenability of fermenting cells to cleavage by AtlA and AM (*Figure 4—figure supplement 1A and B*).

Two models have been proposed to explain the influence of WTA upon AtlA activity. Schlag *et al.* found that the presence of WTA interferes with the binding of AtlA to the cell surface (*Schlag et al., 2010*). Biswas *et al.* found that WTA contributes to proton binding on the cell surface. AtlA activity decreases substantially below pH 6.5 (*Biswas et al., 2012*), and therefore, it was proposed that binding of protons by WTA leads to a decrease in the local pH of the cell surface thereby inhibiting AtlA activity (*Biswas et al., 2012*). We examined the contribution of these two mechanisms in the lysis of fermenting *S. aureus*.

First, the effect of proton binding by WTA upon AtlA activity was examined by decreasing the pH of the murein hydrolase and autolysis assays. We reasoned that an increased concentration of protons would exacerbate the effect of proton binding by WTA. Under this scenario, cells containing an increased abundance of WTA would be expected to be resistant towards AtlA-dependent cleavage at decreased pH. Consistent with this premise, AtlA-dependent lysis of heat-killed WT was dramatically decreased in murein hydrolase assays conducted at a pH of 5 (*Figure 4C* and *Figure 4—figure supplement 1A and B*). Importantly, lysis of fermenting WT cells was still observed while it was nearly absent for those cultured aerobically (*Figure 4C*). In contrast, lysis rates for tunicamycin treated WT were unaltered upon decreasing the pH (*Figure 4C*, *Figure 4—figure supplements 1*

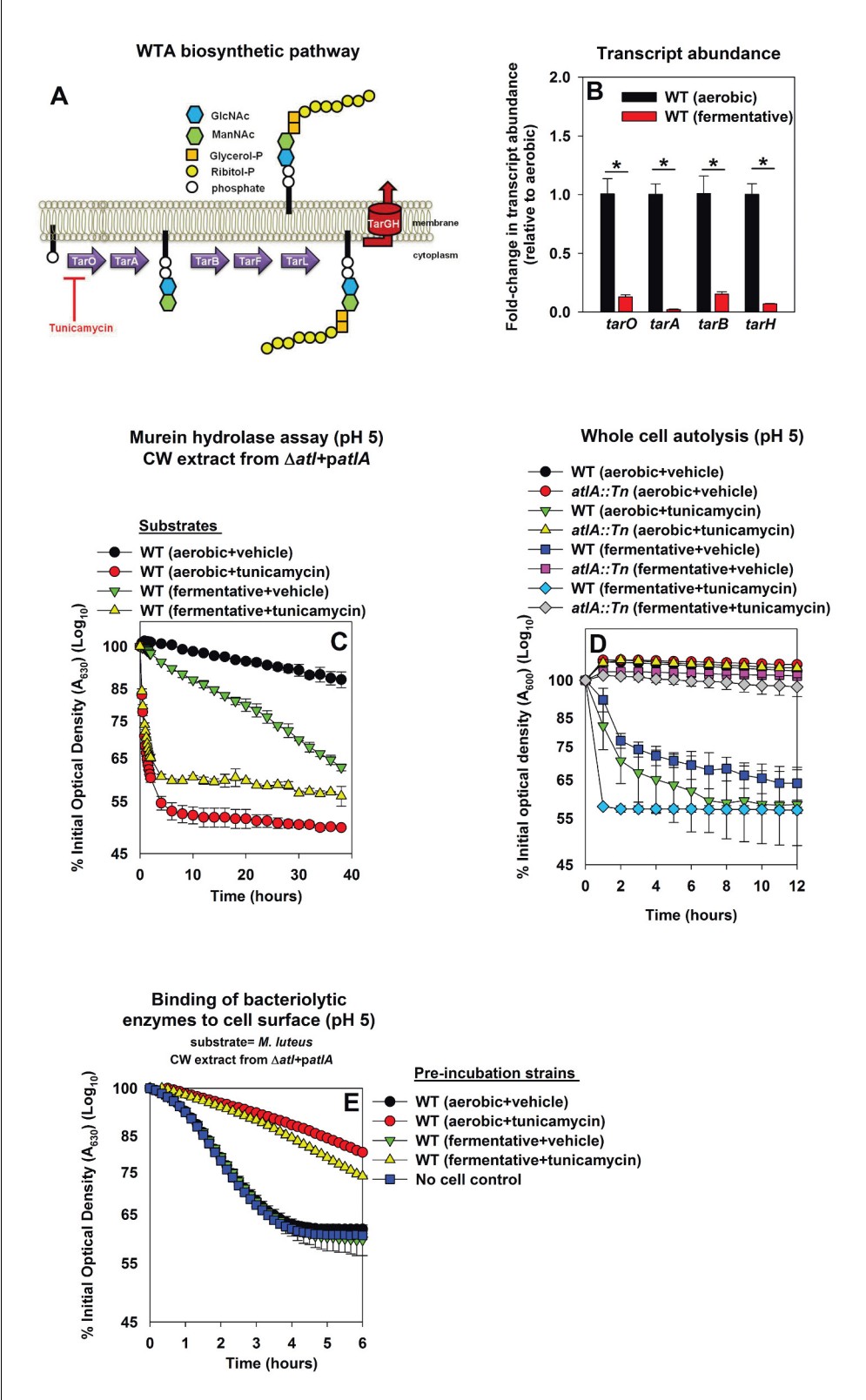

**Figure 4.** Decreased expression of wall-teichoic acids during fermentative growth makes *S. aureus* more amenable to cleavage by AtlA. Panel A; Schematic of wall-teichoic acid (WTA) biosynthesis in *S. aureus*. The diagram displays select proteins involved in WTA biosynthesis and is redrawn as initially presented by *Campbell et al. (2012)*. The initial transformations in the pathway catalyzed by TarO and TarA are non-essential, while the latter steps are essential. Tunicamycin inhibits TarO, as well as the 2-epimerase MnaA, which modulates the substrate levels for TarO (*Campbell et al., 2011*; *Figure 4 continued on next page*

*Figure 4 continued*

*Mann et al., 2016*). MnaA is not displayed. Panel B; Transcript levels corresponding to genes encoding for WTA biosynthesis proteins are decreased upon fermentative growth. Biofilms of the WT (JMB 1100) were cultured aerobically or fermentatively, mRNA was extracted, and the abundances of the *tarO*, *tarA*, *tarB*, and *tarH* transcripts were quantified. The data were normalized to 16S rRNA levels, and thereafter to levels observed aerobically. Panel C; AtlA-dependent cleavage of heat-killed cells at a decreased pH is modulated via wall-teichoic acids. Murein-hydrolase activity at pH of 5 for cell-wall associated proteins (CW-extracts) detached from a *ΔatlA* strain (KB 5000) carrying p*atlA* and incubated with heat-killed cells of the WT cultured aerobically or fermentatively in the presence or absence of 100 ng/mL tunicamycin as substrates is displayed. Panel D; AtlA-dependent autolysis of intact whole cells at decreased pH is modulated via wall-teichoic acids. The WT and *atlA::Tn* (JMB 6625) strains were cultured aerobically or fermentatively in the presence or absence of 100 ng/mL tunicamycin. Autolysis was examined in intact cells resuspended in a buffer with pH of 5. Panels E; Heat-killed aerobic or fermenting WT bind similar amounts of AtlA. CW-extract detached from a *ΔatlA* strain (KB 5000) carrying p*atlA* was incubated at pH 5 with heat-killed WT, cultured aerobically or fermentatively in the presence or absence of 100 ng/mL tunicamycin, or in the absence of cells (control) for 8 min. The cells were separated by centrifugation and bacteriolytic activity in the resultant supernatant was assessed upon heat-killed *M. luteus* as a substrate is displayed. Data in Panel B represents the average value of triplicates. Statistical significance was calculated using a two-tail Student's t-test and * indicates p-value of <0.05. Data in Panels C-E represent the average value of technical duplicates from one set of substrate preparation or autolysis assays. The heat-killed substrates were prepared or autolysis assays were conducted on least three separate occasions and similar results were obtained. Error bars in all panels represent standard deviations. Error bars are displayed for all data, but might be too small to see on occasion.

The following figure supplements are available for figure 4:

**Figure supplement 1.** AtlA- and AM-dependent cleavage of heat-killed cells is modulated via altered expression of wall-teichoic acids.

**Figure supplement 2.** AtlA-dependent lysis rates of heat-killed tunicamycin treated cells are not altered upon alterations in the assay buffer pH.

and *2*), confirming that the influence of pH upon AtlA activity was observed entirely as a result of alterations in WTA expression. The results from autolysis assays conducted at pH 5 lent further support to the findings of the murein hydrolase assays (*Figure 4D*). Strikingly, autolysis was abrogated in aerobically cultured WT, while fermentatively cultured cells or those cultured in the presence of tunicamycin underwent proficient AtlA-dependent autolysis (*Figure 4D*).

Second, we examined whether fermenting WT bind an increased amount of AtlA and whether this is dependent upon WTA expression (*Fournier and Hooper, 2000*). Various heat-killed cells were incubated at pH 5 with CW-extract from a *ΔatlA* strain carrying p*atlA*. The cells were subsequently removed, and the bacteriolytic activity remaining in the supernatants was quantified using heat-killed *M. luteus* cells as substrate. Aerobically or fermentatively cultured heat-killed WT cells did not bind bacteriolytic enzymes while tunicamycin treated cells bound a majority of the bacteriolytic enzymes (*Figure 4E*). We concluded that the complete loss of WTA expression does indeed increase binding of AtlA to the cell surface confirming and extending the findings of *Schlag et al. (2010)*. However, altered AtlA binding to WTA was unlikely to underlie the increased lysis of fermenting cells.

From *Figures 3*, *4*, and *Figure 4—figure supplements 1* and *2*, we concluded that fermenting *S. aureus* had increased expression of AtlA and concomitantly decreased expression of wall-teichoic acids. The combination of these two divergent responses facilitates increased autolysis. Since the changes in expression were accompanied by similar changes in transcription we concluded that impaired respiration elicits programmed cell lysis (PCL).

## Programmed cell lysis and biofilm formation in fermenting cells are governed by the SrrAB two-component regulatory system

Respiration is predominantly mediated by membrane-associated factors. Regulatory system(s) that perceive respiratory status were likely to be membrane-associated. *S. aureus* encodes for 16 two-component regulatory systems (TCRS). Of these, 14 are predicted to employ a membrane-associated histidine kinase. Fermentative biofilm formation was examined in strains that each lacked one individual TCR system (except WalKR, which is essential) (*Dubrac and Msadek, 2004*; *Pang et al., 2014*). A strain lacking the staphylococcal respiratory regulatory system (SrrAB) was attenuated in fermentative biofilm formation (*Figure 5A*). Reintroduction of *srrAB* into the *ΔsrrAB* strain upon an episome restored fermentative biofilm formation (*Figure 5A*). Consistent with SrrAB mediated changes in biofilm formation occurring as a result of altered respiratory status, the introduction of a *ΔsrrAB* mutation into a *hemB::Tn* strain attenuated the increased biofilm formation of the *hemB::Tn*

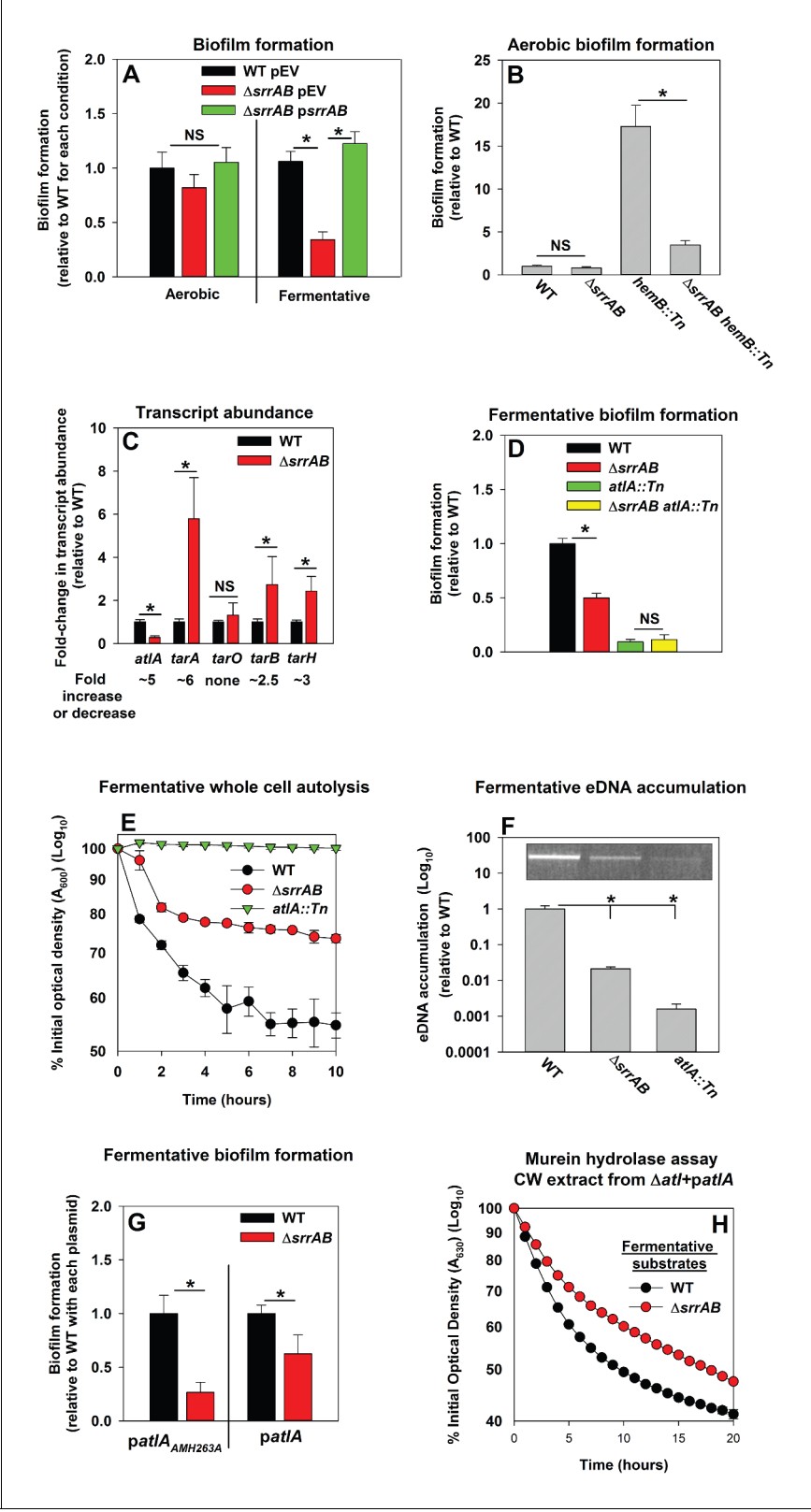

**Figure 5.** Programmed cell lysis and biofilm formation in fermenting cells are governed by the SrrAB two-component regulatory system. Panel A; Fermentative biofilm formation is dependent upon SrrAB. Biofilm formation is displayed following aerobic or fermentative growth in the WT (JMB 1100) carrying pLL39 (pEV) or the *ΔsrrAB* (JMB 1467) strains carrying either pLL39 (pEV) or pLL39_srrAB (p*srrAB*). Panel B; A *hemB* mutant forms SrrAB-dependent biofilms aerobically. Biofilm formation following aerobic growth is displayed for the WT, *ΔsrrAB*, *hemB::Tn* (JMB 6037), and *ΔsrrAB hemB::*
*Figure 5 continued on next page*

*Figure 5 continued*

*Tn* (JMB 6039) strains. Panel C; Transcript levels corresponding to genes involved in programmed cell lysis and biofilm formation are altered in a Δ*srrAB* strain. Biofilms of the WT and Δ*srrAB* strains were cultured fermentatively, mRNA was extracted, and the abundances of the *atlA, tarO, tarA, tarB,* and *tarH* transcripts were quantified. Data were normalized to 16S rRNA levels, and thereafter, to levels observed in the WT. Panel D; The fermentative biofilm formation phenotypes associated with the Δ*srrAB* and *atlA::Tn* mutations are not additive. Biofilm formation is displayed following fermentative growth for the WT, Δ*srrAB, atlA::Tn* (JMB 6625), and Δ*srrAB atlA::Tn* (JMB 6624) strains. Panel E; Autolysis of fermenting *S. aureus* is decreased in a strain lacking SrrAB. The WT, Δ*srrAB,* and *atlA::Tn* strains were cultured fermentatively and autolysis was examined (pH of 5). Panel F; eDNA accumulation is decreased in a strain lacking SrrAB. Biofilms of the WT, Δ*srrAB,* and *atlA::Tn* strains were cultured fermentatively and eDNA was quantified. The data were normalized to the viable cell count and thereafter to the levels in the WT. Panel G; *atlA* in multicopy partially suppresses the biofilm formation defect of the Δ*srrAB* strain. Fermentative biofilm formation is displayed for the WT and Δ*srrAB* strains carrying either p*atlA*$_{AM\ H263A}$ or p*atlA*. Panel H; Heat-killed cells of a Δ*srrAB* strain are less amenable towards AtlA-dependent lysis. Murein-hydrolase activity for cell-wall associated proteins (CW-extracts) detached from a Δ*atlA* strain (KB 5000) carrying p*atlA* and combined with fermentatively cultured and heat-killed WT or Δ*srrAB* strains as substrates are displayed. Data presented represent the average value of eight wells (Panels A, B, D-G) or biological triplicates (Panel C and F). Data in Panels E and H represent the average value of technical duplicates from one set of autolysis assays or substrate preparations. The heat-killed substrates were prepared or autolysis assays were conducted on least three separate occasions and similar results were obtained. Error bars in all panels represent standard deviations. Error bars are displayed for all data, but might be too small to see on occasion. Statistical significance was calculated using a two-tail Student's t-test and p-values>0.05 were considered to be not significant while * indicates p-value of <0.05.

The following figure supplement is available for figure 5:

**Figure supplement 1.** Biofilm formation of a Δ*srrAB* strain is largely unaltered upon supplementing anaerobic biofilms with the alternate terminal electron acceptor nitrate.

strain during aerobic growth (**Figure 5B**). Unlike the WT, anaerobic biofilms formed by the Δ*srrAB* strain were largely unaltered when the growth medium was supplemented with nitrate (**Figure 5— figure supplement 1**).

The influence of SrrAB upon the transcription of genes encoding for factors involved in PCL and biofilm formation was examined. The abundance of the *atlA* transcript was decreased (~5 fold) in the Δ*srrAB* strain (**Figure 5C**). In contrast, the abundances of transcripts corresponding to genes required for WTA biosynthesis were increased in the Δ*srrAB* strain (~2.5–5 fold).

A strain lacking SrrAB displayed phenotypes consistent with decreased expression of AtlA. The fermentative biofilm formation phenotype of the Δ*srrAB atlA::Tn* strain was similar to that of the *atlA::Tn* strain, suggesting that SrrAB influences biofilm formation, in part, via AtlA (**Figure 5D**). Moreover, the Δ*srrAB* strain was deficient in autolysis (**Figure 5E**) and had decreased accumulation of eDNA in its biofilm matrix when cultured fermentatively (**Figure 5F**). To further examine the influence of AtlA upon SrrAB-dependent biofilm formation we introduced multicopy plasmids with alleles encoding for either full length AtlA (p*atlA*) or an enzymatically inactivated AM (p*atlA*$_{AM\ H263A}$) into the Δ*srrAB* strain and examined biofilm formation. The presence of p*atlA* partially suppressed the fermentative biofilm formation defect of the Δ*srrAB* strain when compared to the strain carrying p*atlA*$_{AM\ H263A}$ (**Figure 5G**). Additionally, fermentatively cultured, heat-killed, Δ*srrAB* cells were lysed at a slower rate by CW-extracts from the Δ*atlA* strain carrying p*atlA*, consistent with increased expression of WTA in the Δ*srrAB* strain (**Figure 5H**).

## Genetic evidence suggests that SrrAB-dependent biofilm formation is responsive to the redox status of the menaquinone pool

The cellular molecule(s) that influence SrrAB activity are unidentified. *S. aureus* synthesizes menaquinone and strains lacking menaquinone are unable to respire (**Wakeman et al., 2012**). Upon analyzing previous studies we observed that the transcription of genes positively regulated by SrrAB were reduced in a menaquinone auxotroph (**Kohler et al., 2008**; **Kinkel et al., 2013**; **Yarwood et al., 2001**; **Pragman et al., 2004**). A *hemB* mutant is also unable to respire (**Hammer et al., 2013**) and data presented in **Figure 5B** suggest that SrrAB activity, with respect to biofilm formation, is stimulated in a *hemB::Tn* strain. These seemingly conflicting pieces of information could be readily explained if menaquinone is necessary for SrrAB stimulation.

We reasoned that if SrrAB activity is diminished in the absence of menaquinone then a *hemB::Tn menF::Tn* strain should phenocopy a Δ*srrAB hemB::Tn* strain for biofilm formation. Biofilm formation was examined during aerobic growth in a *hemB::Tn menF::Tn* double mutant, a Δ*srrAB hemB::Tn*

*menF::Tn* triple mutant, as well as their parental strains. The *hemB::Tn* strain displayed increased biofilm formation relative to the *menF::Tn* strain (*Figure 6A*). Importantly, the *ΔsrrAB hemB::Tn*, *hemB::Tn menF:Tn*, and *ΔsrrAB hemB::Tn menF::Tn* strains phenocopied the biofilm formation of the *menF::Tn* strain (*Figure 6A*). These data confirmed that the presence of menaquinone is necessary for SrrAB-dependent biofilm formation in a *hemB::Tn* strain.

Menaquinone functions as both an electron acceptor and an electron donor in the electron transfer chain (ETC) (*Kohler et al., 2008*). Inactivation of heme biosynthesis results in defective terminal oxidases (*Proctor et al., 2006*) and the accumulation of reduced menaquinone. We examined whether a strain enriched for oxidized menaquinone also displayed an increase in the formation of SrrAB-dependent biofilms. *S. aureus* encodes for two NADH::menaquinone oxidoreductases (NdhC and NdhF) and one succinate dehydrogenase (Sdh) (*Schurig-Briccio et al., 2014*; *Gaupp et al., 2010*). A *ΔndhC ndhF::Tn sdh:Tn* strain is deficient in the passage of electrons to menaquinone and consequently enriched in oxidized menaquinone. The *ΔndhC ndhF::Tn sdh:Tn* strain displayed a negligible increase in aerobic biofilm formation (~1.4 fold increase), which was phenocopied by the *ΔsrrAB ΔndhC ndhF::Tn sdh:Tn* strain (*Figure 6B*).

Taken together, the data in *Figure 6* led us to infer that with respect to biofilm formation (1) menaquinone influences SrrAB activity, (2) the absence of menaquinone results in SrrAB being non-responsive, (3) SrrAB activity is increased upon enrichment of reduced menaquinone, and (4) SrrAB is non-responsive to the enrichment of oxidized menaquinone.

## Discussion

Biofilms are the etiologic agents of recurrent staphylococcal infections. Previous work found that hypoxic growth results in increased biofilm formation of *S. aureus*. However, the molecular and regulatory mechanism(s) translating the lack of oxygen into biofilm formation were unknown. We report that oxygen impacts biofilm formation in its capacity as a terminal electron acceptor (TEA) for cellular respiration. Consistent with this premise, supplementing the growth medium with the alternate TEA nitrate decreased biofilm formation during anaerobic growth. Moreover, genetic or chemical inhibition of respiratory processes resulted in increased biofilm formation even in the presence of a TEA. TEA availability in the natural microenvironments of *S. aureus* varies, leading to the supposition that biofilm formation would be responsive to the concentration of TEA. Consistent with this logic, biofilm formation was titratable with respect to the concentration of a TEA or a molecule that inhibits respiration.

Fermenting biofilms were dependent upon the presence of high-molecular weight DNA. High-molecular weight DNA in *S. aureus* biofilm matrixes (eDNA) has been shown to originate from genomic DNA, and thus, its presence suggested that fermenting cells undergo increased autolysis (*Rice et al., 2007*). Lending support to this concept, fermentative biofilm formation was attenuated upon chemical inhibition of autolysis or genetic inactivation of the AtlA murein hydrolase. Fermenting cells underwent increased autolysis in a AtlA-dependent manner and the matrix from the *atlA::Tn* strain had nearly undetectable levels of eDNA. *S. aureus* biofilms incorporate cytosolic proteins into their matrixes and AtlA has been implicated in the release of cytosolic proteins via a process that is not completely understood (*Foulston et al., 2014*; *Pasztor et al., 2010*). We found that fermenting cells had increased activity for a cytosolic protein in the extracellular mileu and an *atlA::Tn* strain was deficient in the release of this protein. Fermenting biofilms were also readily disrupted upon supplementing media with proteinase K suggesting that, in addition to eDNA, proteins are integral components of the fermentative biofilm matrix.

The increased role of AtlA in fermenting biofilms was due to a combination of two divergent cellular responses. First, fermenting cells increased the transcription of *atlA* and autolysis and murein hydrolase assays confirmed that this was translated into increased AtlA activity. Second, fermenting WT cells that had been heat-killed displayed an increased amenability to AtlA-dependent cleavage when used as substrates in murein hydrolase assays. These findings suggested that the cell surface was being altered to facilitate cell lysis. Wall-teichoic acids (WTA) are cell surface glycopolymers that are covalently attached to peptidoglycan and negatively impact AtlA activity (*Biswas et al., 2012*; *Schlag et al., 2010*). The transcription of WTA biosynthesis genes was decreased during fermentative growth. Autolysis and murein hydrolase assays, as well as the WTA synthesis inhibitor tunicamycin, confirmed that WTA expression was decreased during fermentative growth. Since two cellular

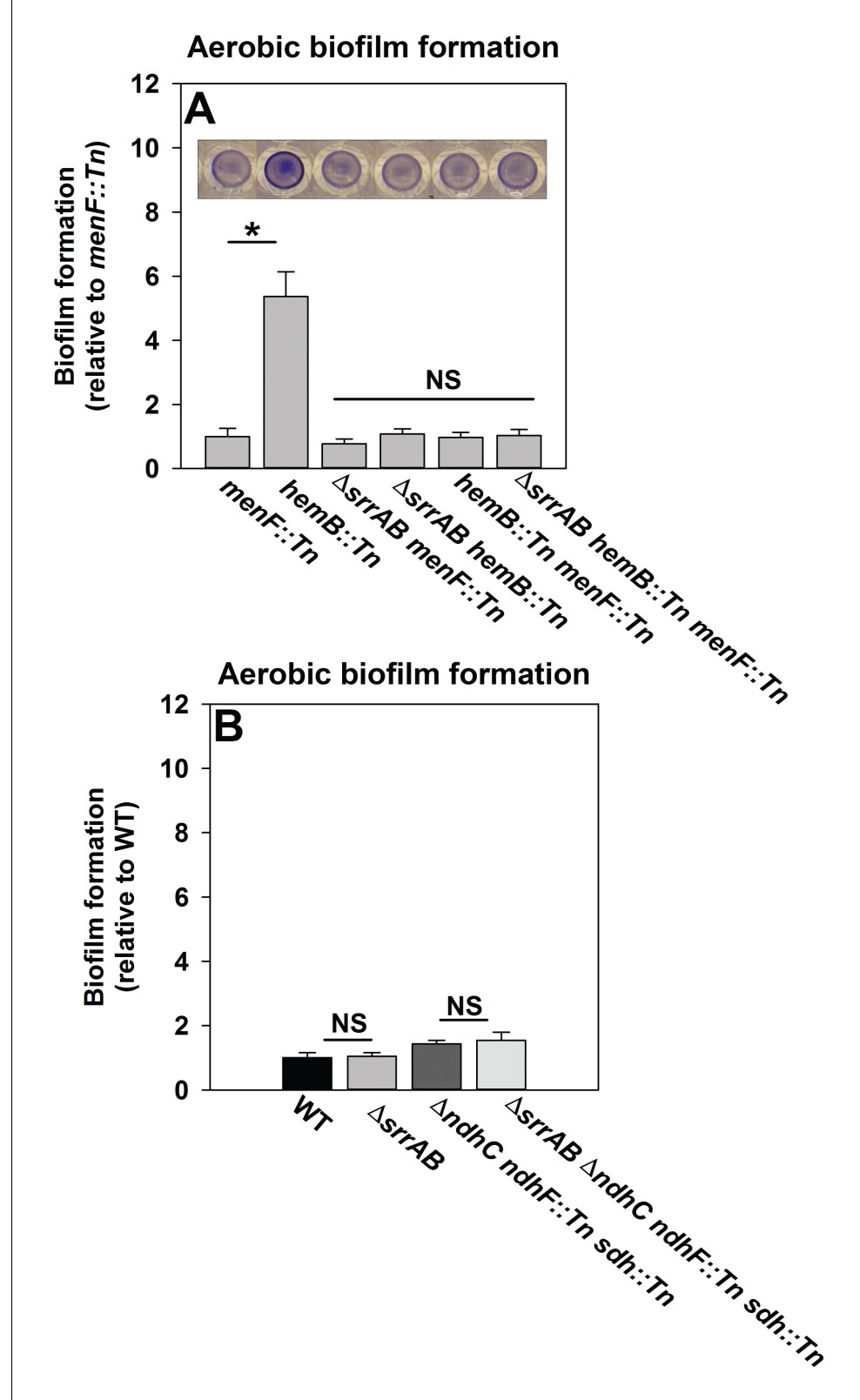

**Figure 6.** SrrAB-dependent biofilm formation is responsive to the oxidation state of the cellular menaquinone pool. Panel A; SrrAB-dependent biofilm formation is inactivated in strains lacking the ability to synthesize menaquinone. Biofilm formation following aerobic growth is displayed for the *menF::Tn* (JMB6219), *hemB::Tn* (JMB6037), *ΔsrrAB menF::Tn* (JMB6221), *ΔsrrAB hemB::Tn* (JMB6039), *hemB::Tn menF::Tn* (JMB6217), and *ΔsrrAB*

*Figure 6 continued on next page*

*Figure 6 continued*

*hemB::Tn menF::Tn* (JMB6673) strains. Panel B; SrrAB-dependent biofilm formation is not stimulated in strains enriched for oxidized menaquinone. Biofilm formation following aerobic growth is displayed for the WT (JMB 1100), Δ*srrAB* (JMB 1467), Δ*ndhC ndhF::Tn sdh:Tn* (JMB 6613), and Δ*srrAB* Δ*ndhC ndhF::Tn sdh:Tn* (JMB 6614) strains. Data in both panels represent the average value of eight wells and the errors bars represent standard deviation. Statistical significance was calculated using a two-tail Student's t-test and p-values>0.05 were considered to be not significant while * indicates p-value of <0.05.

processes are divergently modulated at the transcriptional level in response to an environmental stimulus (TEA availability) to affect autolysis, we propose that this process be termed as programmed cell lysis (PCL), which is illustrated in our working model shown in *Figure 7*.

The cell walls of gram-positive bacteria have been long recognized to serve as proton reservoirs (*Koch, 1986*; *Calamita et al., 2001*). The walls of respiring cells have a low pH and calculations estimate that the local pH can decrease by 3–4 units (*Koch, 1986*; *Calamita et al., 2001*). Further, energy-limiting conditions, such as fermentative growth, or proton trapping, influence bacterial autolysis (*Kemper et al., 1993*; *Jolliffe et al., 1981*). Thus, it has been clear that cell wall composition, the localized pH of the cell wall, and cellular autolysis are interconnected. However, the mechanisms underlying these interconnections have remained elusive. A recent study by Biswas *et al.* shed light on these processes in *S. aureus* (*Biswas et al., 2012*). Biswas *et al.* found that WTA traps protons at the cell surface and they proposed that this results in decreased pH of the microenvironment, and thereby, inhibits AtlA activity (*Biswas et al., 2012*). We found that the influence of pH upon AtlA activity, in both murein hydrolase, as well as autolysis assays, was almost entirely as a result of alterations in WTA expression. These findings both confirm and extend the model put forth by *Biswas et al. (2012)*. An alternate study by Schlag *et al.* proposed that WTA negatively affects AtlA activity by interfering with its binding to the cell surface (*Schlag et al., 2010*). We found that at a pH of 5, tunicamycin treated cells bound a majority of the bacteriolytic activity corresponding to AtlA. In contrast, binding was absent in cells not treated with tunicamycin, regardless of whether they were cultured aerobically or fermentatively. Thus, our findings also confirmed and extended the findings of Schlag *et al.* However, the complete absence of WTA synthesis is unlikely to be a phenomenon that would be physiologically encountered. Therefore, in fermenting *S. aureus*, where the final pH of the culture medium is ~5, we propose that the model of Biswas *et al.* would dominate with respect to autolysis and biofilm formation.

Acidic pH has long been recognized to elicit biofilm formation in *S. aureus* (*Regassa et al., 1992*); however, the mechanisms underlying this phenotype have been unclear. Foulston *et al.* found that cytoplasmic proteins released into the extracellular mileu associate with the exterior of cells, in a pH-dependent and reversible manner, facilitating matrix formation (*Foulston et al., 2014*). The association of the proteins with the cells increases with decreasing pH (*Foulston et al., 2014*). Foulston *et al.* conducted their study in a medium that leads to a decrease in pH over growth (*Foulston et al., 2014*). Thus, it was unclear whether low pH was necessary for the release of the cytoplasmic proteins. Data presented herein suggest that low pH optimizes AtlA function and thereby effects the release of the cytoplasmic proteins, extending the findings of Foulston *et al.* Further, the physiological condition(s) under which this mechanism would be relevant was not entirely clear. In the present study we demonstrate that this mechanism is pertinent in the context of an environmental signal (oxygen) that is crucial in infection progression. Finally, we note that the pH of the skin and nares, which are sites colonized by *S. aureus*, is lower than the homeostatic 7.4 (*Weinrick et al., 2004*). However, to our knowledge, it is unknown if low pH contributes to *S. aureus* biofilm formation *in vivo*.

Respiration is a process mediated predominantly by membrane associated cellular factors. A strain lacking the SrrAB TCRS, consisting of a transmembrane histidine kinase (SrrB) (*Pragman et al., 2004*), was attenuated in biofilm formation. A strain lacking SrrAB had decreased transcription of *atlA*, increased transcription of WTA biosynthesis genes, and displayed multiple phenotypes consistent with the transcriptional data. Further, the biofilm deficient phenotype of the Δ*srrAB* strain was partially suppressed by the introduction of *atlA* in multicopy. These data suggest that SrrAB influences PCL and biofilm formation by divergently influencing AtlA and WTA expression.

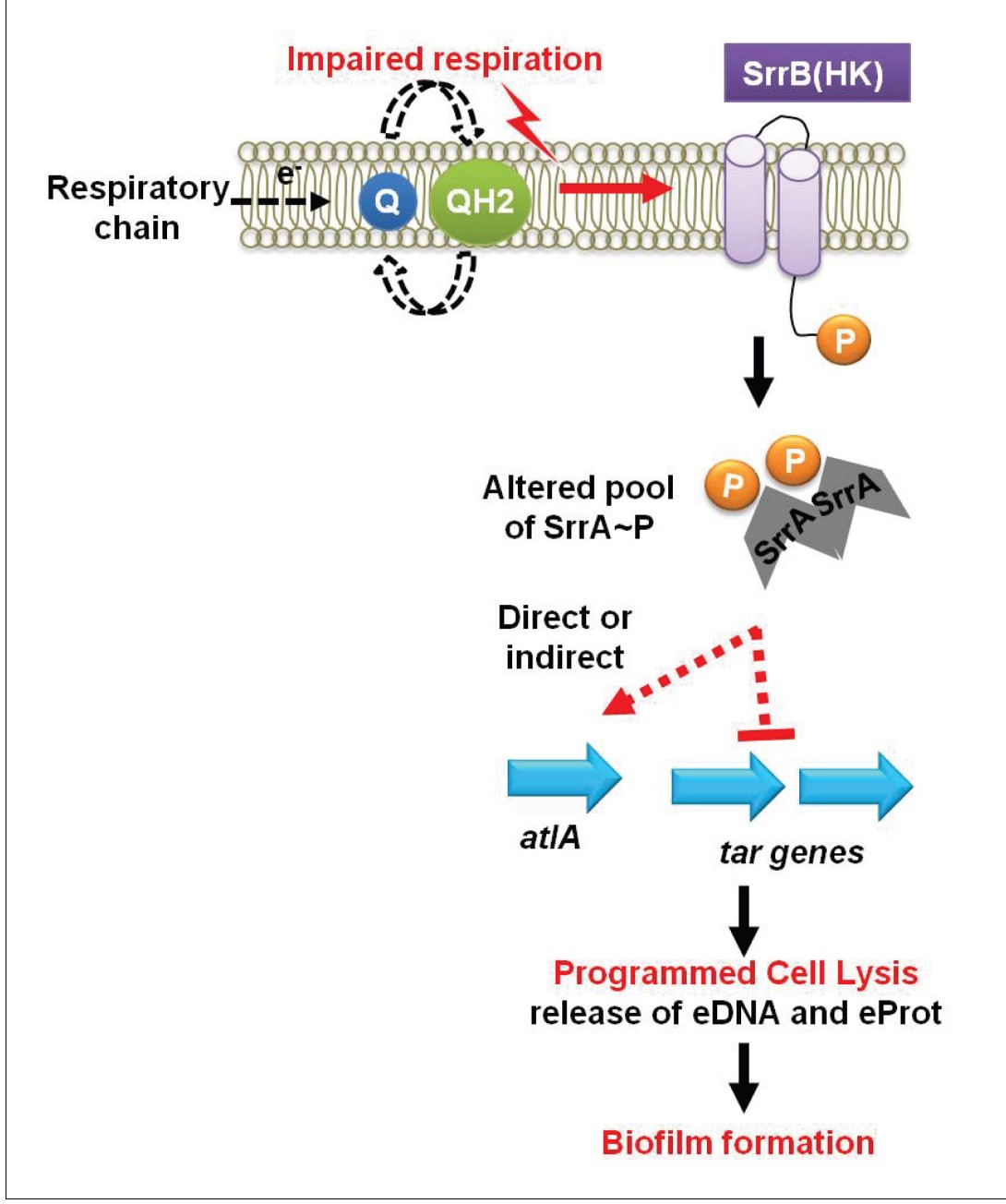

**Figure 7.** A working model for the influence of respiration upon autolysis and biofilm formation in *S. aureus*. A decreased capacity to respire results in an enrichment of reduced menaquinone effecting altered activity of the SrrAB two-component regulatory system. Altered SrrAB activity leads to increased transcription of *atlA* and decreased transcription of genes (*tar*) encoding for wall-teichoic acid (WTA) biosynthesis. The consequent decrease in WTA expression and increase in AtlA expression results in the release of DNA and proteins, cell lysis and biofilm formation. Since cell lysis is effected via regulatory tuning of two divergent processes we term this mechanism as programmed cell lysis (PCL).

SrrAB output was previously shown to be altered under conditions of hypoxia and nitric oxide stress (*Kinkel et al., 2013*). However, the cellular molecule(s) that influence SrrAB activity are unidentified. We found that SrrAB-dependent biofilms increased as a function of decreased respiratory activity. SrrAB-dependent biofilms were formed upon accumulation of reduced, but not oxidized menaquinone, and SrrAB output was abrogated in the absence of menaquinone. These findings

suggest that (1) menaquinone is necessary for stimulus transmission to SrrAB, and (2) the oxidation state of the cellular menaquinone pool influences SrrAB output. We also considered the possibility of two alternate signals that could affect SrrAB output: culture pH and decreased proton motive force. Fermentative growth of *S. aureus* upon TSB results in the release of acidic by-products, which decrease the pH of the extracellular mileu (*Somerville et al., 2003*). Diminished respiration also decreases the proton-motive force. However, heme and menaquinone auxotrophs are both deficient in respiration and the concentration of fermentative by-products and the pH in the spent media is similar in these strains ((*Hammer et al., 2013*) and data not shown). These strains also display a similar decrease in membrane potential (*Hammer et al., 2013*). Yet, only a heme auxotroph forms SrrAB-dependent biofilms. Thus, we deem it unlikely that pH or alterations in proton motive force alter SrrAB activity with respect to biofilm formation.

It is worth noting the similarities that exist between the *Escherichia coli* ArcAB TCRS and SrrAB. Although these TCRS do not display significant homology, the stimuli influencing their activity are similar. ArcB is proposed to donate electrons from conserved cysteine residues to oxidized quinones resulting in silencing of kinase activity (*Malpica et al., 2004*). Similar to ArcB, SrrB contains three conserved cysteine residues, which may facilitate redox interactions with the menaquinone pool. While this leads to the supposition that the molecular mechanism of SrrB signaling may be similar to ArcB, further biochemical analyses are required to make this conclusion. The *Bacillus subtilis* TCRS ResDE displays similarities to SrrAB and it also responds to changes in the respiratory status. However, unlike SrrB, ResE does not contain cysteine residues and studies have deemed it unlikely that the menaquinone pool influences ResDE activity (*Geng et al., 2007*).

Similar to *S. aureus*, *B. subtilis* increases biofilm formation under hypoxic growth and this phenotype is reversed upon supplementation with the alternate TEA nitrate (*Kolodkin-Gal et al., 2013*). Biofilm formation in *B. subtilis* coincided with increased transcription of genes required for matrix production, which was mediated via the membrane-associated kinases KinA and KinB (*Kolodkin-Gal et al., 2013*). *B. subtilis* ResD binds to the promoter regions or within the coding regions of *lytF* and *cwlO*, which encode for two major bacillus autolysins, suggesting it modulates the transcription of these genes (*Henares et al., 2014*; *Ohnishi et al., 1999*; *Ishikawa et al., 1998*; *Yamaguchi et al., 2004*). Further, the binding of ResD to these DNA regions was limited to fermentative growth (*Henares et al., 2014*). However, to our knowledge, it is currently unknown whether ResDE has a role in respiration dependent biofilm formation. The gram-negative bacterium *Pseudomonas aeruginosa* also increases biofilm formation under hypoxic growth and this phenotype is also reversed upon supplementation with the alternate TEA nitrate (*Dietrich et al., 2008*, *2013*). However, the regulatory mechanisms driving respiration dependent biofilm formation in *P. aeruginosa* are unknown. Thus, it seems likely that increased biofilm formation in response to TEA limitation is conserved among diverse bacteria. However, the genetic and regulatory bases underlying biofilm formation may differ.

Clinical isolates of *S. aureus* that are incapable of respiration, termed as small colony variants (SCV), display increased resistance towards antibiotics and cause persistent infections (*Proctor et al., 2006*; *Melter and Radojevič, 2010*). The SCV phenotype often, but not always, arises as a result of mutations in genes necessary for heme biosynthesis resulting in non-functional terminal oxidases (*Hammer et al., 2013*; *Proctor et al., 2006*). Our finding that a heme auxotroph forms SrrAB-dependent biofilms lends considerable insight into the mechanisms that may predominate within clinical SCV strains.

While we suggest the usage of the term PCL in the context of the mechanisms outlined herein, we note that this should not be confused with the process of programmed cell death (PCD) in bacteria or in eukaryotes (*Rice and Bayles, 2008*; *Kerr et al., 1972*; *Kroemer et al., 2009*). Mechanistically, these are distinctly unique processes. Moreover, the morphological and biochemical markers determined in our study do not satisfy the criteria set forth by the committee on cell death (*Kroemer et al., 2009*). However, in the holistic view there are intriguing parallels between *S. aureus* PCL and eukaryotic PCD. PCD occurs as a homeostatic measure in multicellular organisms, whereby a genetically programmed mechanism of cellular catabolism eliminates select quantities and types of cells (*Kerr et al., 1972*; *Kroemer et al., 2009*). PCD is crucial for a variety of processes ranging from proper cell turnover and embryonic development to the functioning of the immune system (*Kerr et al., 1972*; *Kroemer et al., 2009*). While PCD occurs at the cellular level, and typically in a localized environment, it provides benefits at the organismal level (*Kerr et al., 1972*). Similar to

PCD, the findings presented herein suggest that PCL may provide bacteria with a population-level advantage by facilitating biofilm establishment, thereby imparting protection from the immune system and therapeutic agents.

Respiration in eukaryotic cells relies upon using oxygen as a substrate. Similar to PCL, hypoxia or anoxia trigger PCD in eukaryotes (*Shimizu et al., 1996*; *Weinmann et al., 2004*). PCD occurs as one of two distinct biochemical modalities: apoptosis or necrosis. Hypoxia triggered PCD manifests as a mixture of apoptosis and necrosis (*Shimizu et al., 1996*). Anoxia triggered PCD is largely an apoptotic process (*Weinmann et al., 2004*). Interestingly, anoxia-triggered PCD is dependent upon mitochondrial membrane permeabilization by the pro-apoptotic Bcl-2 family proteins Bax and Bak (*Weinmann et al., 2004*; *Kuwana et al., 2002*). Recent evidence suggests that Bax and Bak function as holin-like proteins and facilitate the formation of oligomeric membrane pores (*Kuwana et al., 2002*; *Pang et al., 2011*). *S. aureus* also encodes for two holin-like proteins termed CidA and LrgA (*Ranjit et al., 2011*). The *cid* operon genes, *cidA* and *cidB* have been implicated in programmed cell death in aerobically cultured cells (*Chaudhari et al., 2016*). CidA was previously proposed to have role in cell lysis (*Rice et al., 2007*). This role was predicated upon the phenotype of a *cidA* mutant; however, recent studies suggest that this was likely an outcome of a secondary mutation (*Rice et al., 2007*; *Chaudhari et al., 2016*). CidB, is predicted to be a membrane-associated protein, however its precise function and biochemical activity(s) are yet to be defined (*Rice et al., 2003*; *Windham et al., 2016*; *Chaudhari et al., 2016*). In our hands, *cidA::Tn*, *cidB::Tn*, and *lrgA::Tn* strains were not attenuated in fermentative biofilm formation suggesting a functional separation of the *S. aureus* PCD and PCL pathways, with respect to biofilm formation.

In summary, we report that oxygen impacts *S. aureus* biofilm formation in its capacity as a terminal electron acceptor. Decreased respiration results in programmed cell lysis via increased expression of AtlA and decreased expression of wall-teichoic acids. These processes are governed by the SrrAB TCRS and evidence suggests this occurs in response to the accumulation of reduced menaquinone. The AtlA-dependent release of cytosolic components facilitates biofilm formation.

## Material and methods

### Materials

Restriction enzymes, quick DNA ligase kit, deoxynucleoside triphosphates, and Phusion DNA polymerase were purchased from New England Biolabs. The plasmid mini-prep kit, gel extraction kit and RNA protect were purchased from Qiagen. DNase I was purchased from Ambion. Lysostaphin was purchased from Ambi products. Oligonucleotides were purchased from Integrated DNA Technologies and sequences are listed in *Supplementary file 1*. Trizol and High-Capacity cDNA Reverse Transcription Kits were purchased from Life Technologies. Tryptic Soy broth (TSB) was purchased from MP biomedical. Unless otherwise specified all chemicals were purchased from Sigma-Aldrich and were of the highest purity available.

### Bacterial growth conditions

Overnight cultures of *S. aureus* were grown at 37°C in 10 mL culture tubes containing 1 mL of TSB or 30 mL culture tubes containing 5 mL TSB. Difco BiTek agar was added (15 g $L^{-1}$) for solid medium. When selecting for or against plasmids, antibiotics where added to the following concentrations: 150 µg $mL^{-1}$ ampicillin; 30 µg $mL^{-1}$ chloramphenicol (Cm); 10 µg $mL^{-1}$ erythromycin (Erm); 3 µg $mL^{-1}$ tetracycline (Tet); kanamycin, 125 µg $mL^{-1}$ (Kan); anhydrotetracycline 150 ng $mL^{-1}$.

#### Growth model to assess biofilm formation

Aerobic, overnight cultures, were diluted into fresh TSB and incubated statically at 37°C. For aerobic growth, the cultures were grown in 96-well microtiter plates containing 200 µL in each well or six-well plates containing 6 mL in each well and were covered with an Aera seal (Excel scientific), which allowed for uniform gas exchange. For anaerobic growth, cultures were inoculated aerobically followed immediately by passage through an airlock (three vacuum/gas exchange cycles) into a COY anaerobic chamber equipped with a catalyst to maintain oxygen concentrations below one ppm. Anaerobic growth in the presence of a terminal electron acceptor was achieved by supplementing the media with sodium nitrate (prepared fresh daily).

## Bacterial strains and genetic techniques

Unless otherwise stated, the *S. aureus* strains used in this study (*Table 1*) were constructed in the community-associated *S. aureus* USA300 LAC strain that was cured of the native plasmid pUSA03 that confers erythromycin resistance (*Boles et al., 2010*). Transposon insertions were obtained from the NARSA library that is housed at BEI resources. All *S. aureus* mutant strains and plasmids were verified using PCR, sequencing of PCR products or plasmids (Genewiz, South Plainfield, NJ), or genetic/chemical complementation of phenotypes. *Escherichia coli* DH5α was used as a cloning host for plasmid construction. All constructs were passaged through RN4220 (*Kreiswirth et al., 1983*) and subsequently transduced into the appropriate strains using bacteriophage 80α (*Novick, 1991*).

## Construction of mutant strains and plasmids

The erythromycin resistance cassette in a *menF::Tn* (*ermB*) strain was exchanged to a tetracycline resistance cassette as described earlier, with minor changes (*Bose et al., 2013*). The *menF::Tn* (*ermB*) strain was transduced with the pTnTet plasmid and Tet resistance was selected at 30°C. A single colony was used to inoculate 5 mL of TSB medium and cultured with shaking overnight at 30°C in the presence of Cm. To initiate recombination, cells from the overnight culture were spread onto a TSB agar plate containing Tet and incubated at 42°C (replication non-permissive). Single recombinants were inoculated into 5 mL of TSB and incubated at 30°C in the absence of antibiotic to promote recombination and plasmid loss. These overnights were re-diluted 1:1,000 fold into TSB medium containing 30 ng mL$^{-1}$ of Atet and cultured overnight at 30°C. The overnight culture was diluted of 1:50,000 before plating 20–100 µL onto TSA containing Atet to select against plasmid containing cells. Colonies were screened by replica plating for Cm sensitivity and Tet resistance. The resultant strain, once reconstructed, was verified to be deficient in menaquinone biosynthesis by chemical complementation using menaquinone-4 (MK4). Where mentioned, strains interrupted in *hemB* were verified using chemical complementation by supplementing growth medium with hemin.

The Δ*ndhC::tetM* strain was constructed as described earlier (*Mashruwala et al., 2015*). The pJB38_Δ*srrAB::tet* plasmid was created by using PCR to amplify the *tetM* allele from strain JMB1432 using primers G+tetMluI and G+tetNheI. The PCR product was digested with MluI and NheI and ligated into similarly digested pJB38_Δ*srrAB* (pJB38_Δ*srrAB::tetM*) (*Joska et al., 2014*). The Δ*srrAB::tetM* strain was created as outlined above.

The pLL39_*srrAB* plasmid, containing *srrAB* under the transcriptional control of their native promoter, was constructed using yeast recombinational cloning as previously described (*Joska et al., 2014*; *Mashruwala and Boyd, 2016*; *Mashruwala et al., 2016b*). Amplicons were generated using the following primer pairs: pLL39_yeastF and yeast_srrProR, yeast_srrProF and srrAB_pLL39R. The *srrAB* alleles and the upstream promoter region were amplified from the LAC chromosome and the pLL39 vector was linearized using SalI. The resultant pLL39_*srrAB* plasmid was integrated as an episome into the chromosome of the Δ*srrAB* strain (JMB1467).

## Static model of biofilm formation

Biofilm formation was examined as described earlier, with minor changes (*Mashruwala et al., 2016a*). Overnight cultures were diluted into fresh TSB to a final optical density of 0.05 (A$_{590}$). 200 µL aliquots of diluted cultures were added to the wells of a 96-well microtitre plate (Corning 3268) and the plate was subsequently incubated statically at 37°C for 22 hr. Prior to harvesting the biofilm, the optical density (A$_{590}$) of the cultures was determined. The plate was subsequently washed twice with water, biofilms were heat fixed at 60°C, and the plates were allowed to cool to room temperature. The biofilms were stained with 0.1% crystal violet, washed thrice with water, destained with 33% acetic acid and the absorbance of the resulting solution was recorded at 570 nm, standardized to an acetic acid blank, and subsequently to the optical density of the culture upon harvest. Finally, the data were normalized with respect to the WT or as described in the figure legends to obtain relative biofilm formation.

## Quantitative real-time PCR assays

Biofilms were cultured in the presence or absence of oxygen for eight hours. At point of harvest the spent medium was discarded and the remaining culture was immediately resuspended in RNAProtect reagent (Qiagen) and treated according to manufacturer instructions. The treated culture was

**Table 1.** Strains and plasmids used in this study.

**Strains used in this study**

| S. aureus Strains | Genotype/Description | Genetic Background | Source/Reference |
|---|---|---|---|
| JMB1100 | Wild-type; USA300_LAC (erm sensitive); MRSA; CC8 | LAC | *Boles et al. (2010)* |
| RN4220 | Restriction minus; MSSA; CC8 | NCTC8325 | *Kreiswirth et al. (1983)* |
| JMB 1467 | ΔsrrAB (SAUSA300_1441–42) | LAC | *Pang et al. (2014)* |
| JMB 2047 | ΔsrrAB::tet | LAC | This work |
| JMB 2078 | katA::Tn (ermB) (SAUSA300_1232) | LAC | V. Torres |
| SH1000 | parent; MSSA; CC8 | SH1000 | *Horsburgh et al. (2002)* |
| JMB 1324 | parent, MRSA, USA400, CC1 | MW2 | Alex Horswill and *Centers for Disease Control and Prevention (1999)* |
| JMB 7570 | parent, MRSA, USA100; CC5 | N315 | Ann Stock and *Kuroda et al. (2001)* |
| JMB 1432 | Δfur::tetM | LAC | *Horsburgh et al. (2001)* |
| JMB 6231 | sdhA::Tn(ermB) | LAC | BEI resources and *Fey et al. (2013)* |
| JMB 6232 | ΔsrrAB sdhA::Tn(ermB) | LAC | This work |
| JMB 6384 | ndhF::Tn(ermB) (SAUSA300_0841) | LAC | This work; BEI resources and *Fey et al. (2013)* |
| JMB 2057 | ΔndhC::tet (SAUSA300_0844) | LAC | This work |
| JMB 6614 | ΔsrrAB sdhA::Tn(ermB) ΔndhC::tet ndhF::Tn(ermB) | LAC | This work |
| JMB 6613 | sdhA::Tn(ermB) ΔndhC::tet ndhF::Tn(ermB) | LAC | This work |
| JMB 6037 | hemB::Tn(ermB) | LAC | BEI resources and *Fey et al. (2013)* |
| JMB 6039 | ΔsrrAB hemB::Tn(ermB) | LAC | This work |
| JMB 6029 | menF::Tn(ermB) | LAC | BEI resources and *Fey et al. (2013)* |
| JMB 6033 | ΔsrrAB menF::Tn(ermB) | LAC | This work |
| JMB 6219 | menF::Tn(tet) | LAC | This work |
| JMB 6221 | ΔsrrAB menF::Tn(tet) | LAC | This work |
| JMB 6217 | hemB::Tn(ermB) menF::Tn(tet) | LAC | This work |
| JMB 6673 | ΔsrrAB hemB::Tn(ermB) menF::Tn(tet) | LAC | This work |
| JMB 6625 | atlA::Tn(ermB) | LAC | BEI resources and *Fey et al. (2013)* |
| KB5000 | ΔatlA | UAMS-1 | *Bose et al. (2012)* |
| JMB 6624 | ΔsrrAB atlA::Tn(ermB) | LAC | This work |
| JMB 5577 | icaA::Tn(ermB) | LAC | This work; BEI resources and *Fey et al. (2013)* |
| JMB 5579 | icaB::Tn(ermB) | LAC | This work; BEI resources and *Fey et al. (2013)* |
| JMB 5578 | icaC::Tn(ermB) | LAC | This work; BEI resources and *Fey et al. (2013)* |
| JMB 7270 | hmrA::Tn(ermB) | JE2 | BEI resources and *Fey et al. (2013)* |
| JMB 7265 | lytN::Tn(ermB) | JE2 | BEI resources and *Fey et al. (2013)* |
| JMB 7267 | lytX::Tn(ermB) | JE2 | BEI resources and *Fey et al. (2013)* |
| JMB 7266 | sle1::Tn(ermB) | JE2 | BEI resources and *Fey et al. (2013)* |
| JMB 7268 | lytY::Tn(ermB) | JE2 | BEI resources and *Fey et al. (2013)* |
| JMB 7269 | lytZ::Tn(ermB) | JE2 | BEI resources and *Fey et al. (2013)* |
| JMB 7271 | lytM::Tn(ermB) | JE2 | BEI resources and *Fey et al. (2013)* |
| JMB2977 | parent | JE2 | BEI resources and *Fey et al. (2013)* |
| JMB7277 | narG::Tn (ermB) | LAC | BEI resources and *Fey et al. (2013)* |
| JMB 1148 | ΔhptRS | LAC | *Pang et al. (2014)* |
| JMB 1357 | ΔlytSR | LAC | *Pang et al. (2014)* |
| JMB 1330 | graS::erm | LAC | *Boles et al. (2010)* |

*Table 1 continued on next page*

*Table 1 continued*

**Strains used in this study**

| S. aureus Strains | Genotype/Description | Genetic Background | Source/Reference |
|---|---|---|---|
| JMB 1335 | ΔsaePQRS::spec | LAC | *Nygaard et al. (2010)* |
| JMB 1219 | ΔSAUSA300_1219–1220 | LAC | *Pang et al. (2014)* |
| JMB 1383 | ΔarlSR | LAC | *Pang et al. (2014)* |
| JMB 1358 | ΔphoSR | LAC | *Pang et al. (2014)* |
| JMB 1241 | ΔairSR | LAC | *Pang et al. (2014)* |
| JMB 1377 | ΔvraSR | LAC | *Pang et al. (2014)* |
| JMB 1333 | Δagr::tetM | LAC | *Kiedrowski et al. (2011)* |
| JMB 1223 | ΔkdpSR | LAC | *Pang et al. (2014)* |
| JMB 1359 | ΔhssSR | LAC | *Pang et al. (2014)* |
| JMB 1145 | ΔnreSR | LAC | *Pang et al. (2014)* |
| JMB 1232 | ΔSAUSA300_2558–2559 | LAC | *Pang et al. (2014)* |

**Other Strains**

*Escherichia coli* PX5

*Sacchromyces cerevisiae* FY2

**Plasmids used in this study**

| Plasmid name | Insert Locus/function | Source/Reference |
|---|---|---|
| pJB38 | Insertless vector for cloning chromosomal gene deletions | *Bose et al. (2013)* |
| pJB38_srrAB::tet | Construction of srrAB::tet allele | This work |
| pCM28 | Insertless cloning vector | A. Horswill |
| pCM28_srrAB | srrAB complementing vector | *Mashruwala and Boyd (2017)* |
| pLL39 | Insertless cloning vector for genetic complementation | *Luong and Lee (2007)* |
| pLL39_srrAB | srrAB complementing vector | This work |
| pJB141 | atlA complementing vector | *Bose et al. (2012)* |
| pJB135 | atlA_GL complementing vector | *Bose et al. (2012)* |
| pJB122 | atlA_AMH263A complementing vector | *Bose et al. (2012)* |
| pJB128 | Insertless cloning vector | *Bose et al. (2012)* |
| pJB111 | atlA_AM complementing vector | *Bose et al. (2012)* |
| pTnTet | Construction of menF::Tn (Tet) | *Bose et al. (2013)* |

subjected to centrifugation, the supernatant was discarded, and the cell pellet was resuspended in RNase free 50 mM Tris, pH 8. Cell-free extracts were generated using bead beating. RNA was extracted using Trizol, as per manufacturer instructions. Downstream treatments of the purified RNA and construction of cDNA libraries was as described earlier (*Mashruwala et al., 2015*). Primers for PCR were designed manually or using the Primer Express 3.0 software from Applied Biosystems. Quantitative real time PCR reactions (Table S1) were conducted as described earlier (*Mashruwala et al., 2015*).

## Quantification of high-molecular weight extracellular DNA (eDNA)

eDNA was analyzed as described earlier with some changes (*Kaplan et al., 2012*). Overnight cultures were diluted into TSB to a final optical density of 0.05 ($A_{600}$) in a final volume of 6 mL per well of a six-well plate. The cultures were incubated statically at 37°C for 22 hr. At point of harvest, the spent media supernatant was aspirated out of each well. One mL of 1X phosphate buffered saline

(PBS) was immediately added to the wells and a cell scraper was used to transfer the contents to an eppendorf tube. The biomass was pelleted by centrifugation and the supernatant was removed by aspiration. The pellets were thoroughly resuspended in 1X PBS and vortexed for 5 min using a Vortex Genie 2 (Scientific Industries) at the highest speed possible using a vertical micro-tube adapter. Aliquots were removed for determination of the viable cell count (colony forming units) and samples were pelleted by centrifugation. Control experiments verified that the viable cell counts were not affected by the vortexing procedure (data not shown). Equal volumes of the supernatants were assessed for the presence of high molecular weight DNA (>10 kilobases) using agarose gel electrophoresis. To assess the extracellular DNA in a semi-quantitative manner, the gels were photographed and the bands were subjected to density analysis using Image J software. For each sample, the spot densities were normalized to the viable cell count (colony forming units) and subsequently as mentioned in the figure legends.

## Cytoplasmic protein release assays

Strains were cultured as described under eDNA analyses. The samples were vortexed briefly, biomass was transferred into a microcentrifuge tube, and cell pellets and spent media supernatants were partitioned by centrifugation. The spent media supernatant was retained for further analyses. The cell pellets were resuspended in lysis buffer (50 mM Tris, 150 mM NaCl, 4 µg lysostaphin, 8 µg DNAse, pH 7.5) and incubated at 37°C until confluent lysis was observed. Cell lysates were clarified using centrifugation to obtain cell-free extracts. Catalase (Kat) activity was assayed, in both the cell-free extracts as well as spent medium supernatants as described elsewhere (*Mashruwala et al., 2016a*; *Beers and Sizer, 1952*). The ratio of extracellular to intracellular Kat activity was utilized to determine protein release. In control experiments, Kat activity was undetectable in a *katA::Tn* strain (data not shown).

## Whole cell autolysis assays

Overnight cultures were diluted into TSB to a final optical density of 0.05 ($A_{600}$) and cultured for four hours. Whole cell autolysis assays were conducted as described elsewhere with minor changes (*Bose et al., 2012*). Briefly, the cultures were harvested by centrifugation, cell pellets were washed twice, and resuspended in autolysis buffer (50 mM HEPES, 150 mM NaCl, 0.05% Triton X-100, pH 7.5). For analyses conducted at pH 5, HEPES was replaced with 0.2 M sodium acetate buffer and all other components remained unaltered. The cell suspensions were then incubated at 37°C with shaking and optical densities were recorded periodically.

## Murein hydrolase assays

Biofilms were cultured for four hours and cells were harvested as mentioned under eDNA analyses. Thereafter, cell-wall associated protein extracts (CW-extracts) were prepared and murein hydrolase activity determined as described elsewhere with minor changes (*Mani et al., 1993*). Briefly, cell pellets were washed and CW-extracts were prepared by resuspension in 3 M lithium chloride and incubation for 25 min (*Mani et al., 1993*). Protein concentrations of the extracts were determined and between 0.1–0.5 µg of an individual extract was combined with heat-killed cell substrates (0.35 optical density ($A_{600}$)) in assay buffer (50 mM Hepes, 150 mM NaCl, 0.01% Triton X-100, pH 7.5). For analyses conducted at pH 5, HEPES was replaced with 0.2 M sodium acetate and all other components remained unaltered. Samples were incubated with shaking at 37°C and optical densities were recorded periodically. Binding assays were conducted as earlier (*Fournier and Hooper, 2000*).

## Acknowledgements

The Boyd lab is supported by Rutgers University, the Charles and Johanna Busch foundation and USDA MRF project NE−1028. AAM is supported by the Douglas Eveleigh fellowship from the Microbial Biology Graduate Program and an Excellence Fellowship from Rutgers University. The authors would like to thank Dr. William Belden for use of his real-time thermocycler. We thank Dr. Jeffrey Bose and Dr. Kenneth Bayles for kindly sharing the *atlA* plasmids and strains with us. We thank Dr. Alex Horswill and Dr. Ann Stock for sharing *S. aureus* clinical isolates.

## Additional information

### Funding

| Funder | Grant reference number | Author |
| --- | --- | --- |
| U.S. Department of Agriculture | multistate project NE1048 | Jeffrey M Boyd |
| Rutgers University Busch Bio-medical Grant | | Jeffrey M Boyd |

The funders had no role in study design, data collection and interpretation, or the decision to submit the work for publication.

### Author contributions

AAM, Conceptualization, Data curation, Formal analysis, Supervision, Investigation, Visualization, Methodology, Writing—original draft, Writing—review and editing; AvdG, Validation, Investigation, Writing—review and editing; JMB, Conceptualization, Resources, Supervision, Funding acquisition, Project administration, Writing—review and editing

### Author ORCIDs

Ameya A Mashruwala, http://orcid.org/0000-0001-5583-4174
Adriana van de Guchte, http://orcid.org/0000-0002-0771-3842
Jeffrey M Boyd, http://orcid.org/0000-0001-7721-3926

## Additional files

### Supplementary files

• Supplementary file 1. Oligonucleotides used in this study for real-time quantitative PCR and cloning.

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
