## [Decision Letter]

Thank you for submitting your article "Impaired respiration elicits SrrAB-dependent programmed cell lysis and biofilm formation in *Staphylococcus aureus*" for consideration by *eLife*. Your article has been favorably evaluated by Wendy Garrett (Senior Editor) and three reviewers, one of whom, Michael S Gilmore (Reviewer #1), is a member of our Board of Reviewing Editors.

The reviewers have discussed the reviews with one another and the Reviewing Editor has drafted this decision to help you prepare a revised submission.

Summary:

This is a well-done investigation into the relationship between anoxia and biofilm formation by *S. aureus*. This work is most notable for the effort undertaken to independently validate observations by attacking the problem from several perspectives. In general, insights are initially gained into biofilm formation using genetic tools. These are systematically followed by examining in depth the connection between the genetics associated with increased or decreased biofilm formation in response to reduced oxygen availability, and predicted behaviors tested in other ways. Although there are few surprises and much of the process is what would be expected based on the prior work of others, biofilm formation is complex and the field has been muddied by superficial studies that have drawn sweeping conclusions based on dogma and limited testing. A main virtue of this work is that it is complete, systematic and rigorous, and most likely applies to biofilm formation under most conditions and by most strains relevant to infection.

In contrast to the strong technical section of this manuscript, the Introduction is comparatively superficial and uncritically reiterates much of the dogma in the field. It would be enhanced and strengthened by examining the factual basis for many statements. For example, although a cause of infective endocarditis, this is hardly the most common manifestation of *S. aureus* infection, or the most likely to lead to serious consequence. It therefore is not a good choice as representative for framing the importance of the work. Emerging resistance to last line antibiotics would be better to include linezolid and daptomycin – frank resistance to vancomycin remains rare, although the utility of vancomycin is often compromised by the "intermediate" susceptibility of VISA strains. The statement "A majority of staphylococcal infections are intimately connected to the ability of the bacterium to form complex multicellular communities called biofilms…" cites work that made similar statements, but did not critically examine the question, and hence makes a thinly supported case, etc. This is not to say that a strong case cannot be made, but as written, it is not convincing.

The technical meat and potatoes of the paper are strong. *S. aureus* biofilm formation is unambiguously greater in reduced oxygen, and by probing this with the alternate terminal electron acceptor nitrate, and showing the critical role of heme in relaying electrons along the partial electron transport pathway of *S. aureus*, the authors make a strong case for the its importance. One area that is less explored/discussed are the alternate fates of electrons in forme fruste respiration in impaired mutants, and precisely how that may limit biofilm formation. This is indirectly pursued by showing an effect on expression on transcription of LTA biosynthetic genes, and more directly using tunicamycin to inhibit LTA biosynthesis, but tunicamycin also inhibits targets other than TarO. The authors complement this by examining the effect of deficiencies in LTA production with alteration of AtlA binding and activity, which is nicely done. This raises the questions, what is known about the pH where biofilms are problematic, and to what degree is this consistent with the mechanism proposed by the investigators? Given the role of respiration energetics in events leading to biofilm formation, it was then not surprising to find the SrrAB TCS, or menaquinones involved. Nevertheless, these lines of experimentation generated results consistent with the model and add further support and rigor to the manuscript.

In summary, this is a technically well done study that adds considerable clarity to an otherwise congested field that in this reviewer's opinion has been confounded by the dogma and lore of biofilms developed largely in other systems. For example, production of specific extracellular polymers, such as PIA, fit expectations for biofilm formation based on what had been discovered in other systems, but as the authors and others have shown, are not likely the most important element of biofilm formation by most clinical isolates of *S. aureus*. Some reviewers expressed skepticism about connecting the mechanism identified here to other types of processes, as may be superficially inferred by dubbing this process "programmed etc..…", as it may have the unintended consequence of building in expectations developed in the eukaryotic programmed cell death/tissue differentiation paradigm, which may or may not apply to biofilm formation by *S. aureus*. Irrespective of how the authors proceed on that, as they are not the first to venture in that direction, they should add a few sentences to help the reader reconcile their model with other "programmed cell death"-type models that have been published.

Essential revisions:

1) Some reconciliation of "Programmed" models as noted above is needed in the Discussion.

2) Provide details on fermentation conditions to inform the reader as to whether or how available oxygen was measured or inferred (e.g., were the experiments performed in anaerobic environmental chambers?)

---

## [Author Response]

*In contrast to the strong technical section of this manuscript, the Introduction is comparatively superficial and uncritically reiterates much of the dogma in the field. It would be enhanced and strengthened by examining the factual basis for many statements. For example, although a cause of infective endocarditis, this is hardly the most common manifestation of S. aureus infection, or the most likely to lead to serious consequence. It therefore is not a good choice as representative for framing the importance of the work. Emerging resistance to last line antibiotics would be better to include linezolid and daptomycin – frank resistance to vancomycin remains rare, although the utility of vancomycin is often compromised by the "intermediate" susceptibility of VISA strains. The statement "A majority of staphylococcal infections are intimately connected to the ability of the bacterium to form complex multicellular communities called biofilms…" cites work that made similar statements, but did not critically examine the question, and hence makes a thinly supported case, etc. This is not to say that a strong case cannot be made, but as written, it is not convincing.*

We thank the referee for their points and we agree with them. We have expanded the Introduction in the revised manuscript to address these points.

*The technical meat and potatoes of the paper are strong. S. aureus biofilm formation is unambiguously greater in reduced oxygen, and by probing this with the alternate terminal electron acceptor nitrate, and showing the critical role of heme in relaying electrons along the partial electron transport pathway of S. aureus, the authors make a strong case for the its importance. One area that is less explored/discussed are the alternate fates of electrons in forme fruste respiration in impaired mutants, and precisely how that may limit biofilm formation. This is indirectly pursued by showing an effect on expression on transcription of LTA biosynthetic genes, and more directly using tunicamycin to inhibit LTA biosynthesis, but tunicamycin also inhibits targets other than TarO. The authors complement this by examining the effect of deficiencies in LTA production with alteration of AtlA binding and activity, which is nicely done. This raises the questions, what is known about the pH where biofilms are problematic, and to what degree is this consistent with the mechanism proposed by the investigators?*

We thank the reviewer for raising this point. We have included text in the Discussion of the revised manuscript to address this point.

*Given the role of respiration energetics in events leading to biofilm formation, it was then not surprising to find the SrrAB TCS, or menaquinones involved. Nevertheless, these lines of experimentation generated results consistent with the model and add further support and rigor to the manuscript.*

*In summary, this is a technically well done study that adds considerable clarity to an otherwise congested field that in this reviewer's opinion has been confounded by the dogma and lore of biofilms developed largely in other systems. For example, production of specific extracellular polymers, such as PIA, fit expectations for biofilm formation based on what had been discovered in other systems, but as the authors and others have shown, are not likely the most important element of biofilm formation by most clinical isolates of S. aureus. Some reviewers expressed skepticism about connecting the mechanism identified here to other types of processes, as may be superficially inferred by dubbing this process "programmed etc..…", as it may have the unintended consequence of building in expectations developed in the eukaryotic programmed cell death/tissue differentiation paradigm, which may or may not apply to biofilm formation by S. aureus. Irrespective of how the authors proceed on that, as they are not the first to venture in that direction, they should add a few sentences to help the reader reconcile their model with other "programmed cell death"-type models that have been published.*

*Essential revisions:*

*1) Some reconciliation of "Programmed" models as noted above is needed in the Discussion.*

We agree with the referee and editors that our usage of programmed cell lysis (PCL) may be confused with programmed cell death (PCD). We have added text to the discussion to clarify this point. We also discuss some of the similarities, from a holistic point of view, that exist between eukaryotic PCD and *S. aureus* PCL.

*2) Provide details on fermentation conditions to inform the reader as to whether or how available oxygen was measured or inferred (e.g., were the experiments performed in anaerobic environmental chambers?)*

Thank you for noting this. We used an anaerobic chamber to conduct all of our anaerobic experiments. We have included text to clarify this in the Results section corresponding to Figure 1 of the revised manuscript. Moreover, precise details on the number of vacuum/gas exchange cycles and conditions under which cultures are inoculated are provided under the growth conditions in the Methods section of the revised manuscript.